# Information transfer in mammalian glycan-based communication

**Felix F Fuchsberger**[1,2,3†], **Dongyoon Kim**[1,2,3†], **Natalia Baranova**[1,3], **Hanka Vrban**[1,3], **Marten Kagelmacher**[2], **Robert Wawrzinek**[1,2,3], **Christoph Rademacher**[1,2,3]*

[1]Department of Pharmaceutical Sciences, University of Vienna, Vienna, Austria; [2]Department of Biomolecular Systems, Max Planck Institute of Colloids and Interfaces, Potsdam, Germany; [3]Department of Microbiology, Immunology and Genetics University of Vienna, Max F. Perutz Labs, Vienna, Austria

**\*For correspondence:**
Christoph.rademacher@univie.
ac.at

[†]These authors contributed
equally to this work

**Competing interest:** The authors
declare that no competing
interests exist.

**Reviewing Editor:** Andre
Levchenko, Yale University,
United States

**Abstract** Glycan-binding proteins, so-called lectins, are exposed on mammalian cell surfaces and decipher the information encoded within glycans translating it into biochemical signal transduction pathways in the cell. These glycan-lectin communication pathways are complex and difficult to analyze. However, quantitative data with single-cell resolution provide means to disentangle the associated signaling cascades. We chose C-type lectin receptors (CTLs) expressed on immune cells as a model system to study their capacity to transmit information encoded in glycans of incoming particles. In particular, we used nuclear factor kappa-B-reporter cell lines expressing DC-specific ICAM-3–grabbing nonintegrin (DC-SIGN), macrophage C-type lectin (MCL), dectin-1, dectin-2, and macrophage-inducible C-type lectin (MINCLE), as well as TNFαR and TLR-1&2 in monocytic cell lines and compared their transmission of glycan-encoded information. All receptors transmit information with similar signaling capacity, except dectin-2. This lectin was identified to be less efficient in information transmission compared to the other CTLs, and even when the sensitivity of the dectin-2 pathway was enhanced by overexpression of its co-receptor FcRγ, its transmitted information was not. Next, we expanded our investigation toward the integration of multiple signal transduction pathways including synergistic lectins, which is crucial during pathogen recognition. We show how the signaling capacity of lectin receptors using a similar signal transduction pathway (dectin-1 and dectin-2) is being integrated by compromising between the lectins. In contrast, co-expression of MCL synergistically enhanced the dectin-2 signaling capacity, particularly at low-glycan stimulant concentration. By using dectin-2 and other lectins as examples, we demonstrate how signaling capacity of dectin-2 is modulated in the presence of other lectins, and therefore, the findings provide insight into how immune cells translate glycan information using multivalent interactions.

## Editor's evaluation

This manuscript lays out the framework for addressing an important challenge in our understanding of cellular signal transduction: how complex extracellular inputs can be detected and processed using multiple receptors. This problem is addressed in the context of glycan receptors lectins, mediating very common but still not completely understood cell-cell interactions. Using information capacity analysis, the study addresses the importance of glycan input measurement by multiple receptors on the immune cells, showing how the signal detection can benefit from receptor crosstalk.

## Introduction

Glycans are present in all living cells and play a key role in many essential biological processes including development, differentiation, and immunity. Being surface exposed, glycans often encode for information in cellular communication such as self-/non-self-discrimination, cellular identity, and homing as well as apoptosis markers (*Bode et al., 2019*; *Maverakis et al., 2015*; *Williams, 2017*). Other than linear biopolymers, such as proteins and nucleic acids, glycans are branched structures, where subtle changes in the glycosidic bonds between each monomer can carry essential pieces of information. Adding to this complexity, glycans are products of large cellular machinery and are therefore not directly encoded by the genome (*Cummings, 2009*). Besides their composition, the recognition of glycans by their receptors is complicated, particularly due to the lack of specificity. Glycans are recognized by lectins, yet no glycan is recognized by a single receptor, and no individual lectin is highly specific for only one glycan. Additionally, affinities are low, and interactions often depend on the multivalency of both the receptor and the ligand. Overall, since alterations of the glycocalyx do not function as a deterministic on/off switch but rather a progressive tuning of the cellular response, glycan lectin communication should be considered as a stochastically behaving system, rather than a deterministic one (*Dennis, 2015*).

Many lectin receptors serve as triggers for multiple immunological signaling pathways, often funneling down to NF-κB (nuclear factor kappa-B) as a transcription factor. In this work, we focus on C-type lectin receptors (CTLs). MINCLE (macrophage-inducible C-type lectin), for example, is a CTL involved in the recognition of pathogens as well as self-damage (*Miyake et al., 2015*; *Williams, 2017*). MINCLE and its close relative dectin-2 (dendritic cell-associated C-type lectin-2) signal via the FcRγ gamma chain (*Miyake et al., 2015*; *Ostrop et al., 2015*; *Sato et al., 2006*), leading to CARD9-BCL-10-Malt1 activation (*Figure 1A*). This in turn results in the activation of NF-κB, eventually triggering cytokine release. Importantly, these two receptors share the same signal transduction pathway, while having different functions (*Thompson et al., 2021*). Therefore, both dectin-2 and MINCLE can be compared of whether these related proteins differently transmit glycan information from the receptor level. In contrast, dectin-1 and dectin-2 have different signal transduction pathways but are both involved in the detection of β-glucans and mannan, respectively (*Figure 1A*).

Upon fungal infection, combination of these and other cell surface receptors expressed by antigen presenting cells then leads to a defined immune reaction via signal integration processes (*Snarr et al., 2017*). Such signal integration can result in synergism between the receptors triggering an effect greater than their individual contributions (*Ostrop and Lang, 2017*). For example, MCL (macrophage C-type lectin), another CTL present on cells of the innate immune system, is known to synergistically work with dectin-2 (*Ostrop et al., 2015*; *Zhu et al., 2013*). Additionally, to this type of synergism, other members of the CTL family, e.g., DC-specific ICAM-3–grabbing nonintegrin (DC-SIGN) and Langerin, rather modulate a response instead of initiating it by themselves (*Geijtenbeek and Gringhuis, 2016*; *Osorio and Reis e Sousa, 2011*). Therefore, it is important to quantitatively account for the resulting signaling to describe the complexity of how these cell surface receptors can modulate each other to translate a glycan-encoded information into a biological response.

Accounting for the stochastic behavior of cellular signaling, information theory provides robust and quantitative tools to analyze complex communication channels. A fundamental metric of information theory is entropy, which determines the amount of disorder or uncertainty of variables. In this respect, cellular signaling pathways having high variability of the initiating input signals (e.g. stimulants) and the corresponding highly variable output response (i.e. cellular signaling) can be characterized as a high entropy. Importantly, input and output can have mutual dependence, and therefore, knowing the input distribution can partly provide the information of output distribution. If noise is present in the communication channel, input and output have reduced mutual dependence. This mutual dependence between input and output is called mutual information. Mutual information is, therefore, a function of input distribution, and the upper bound of mutual information is called channel capacity (Appendix 2; *Cover and Thomas, 2012*).

In this report, a communication channel describes signal transduction pathway of CTL, which ultimately lead to NF-κB translocation and finally GFP expression in the reporter model (*Figure 1A*). To quantify the signaling information of the communication channels, we used channel capacity. Importantly, the channel capacity is not merely describing the resulting maximum intensity of the reporter

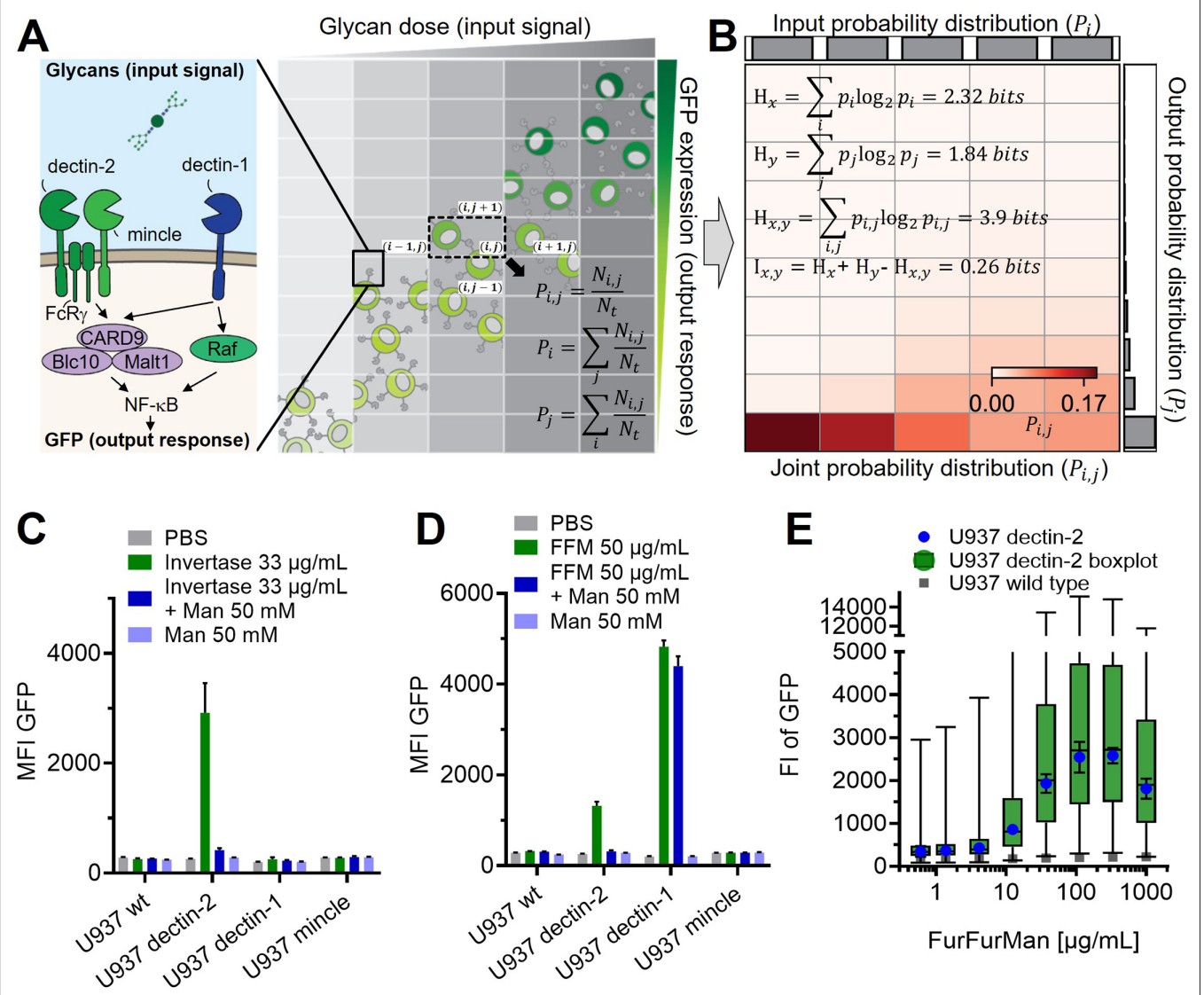

**Figure 1.** Reporter cell system for the observation of glycan-lectin interactions. (**A**) Schematic representation of dectin-1, dectin-2, and macrophage-inducible C-type lectin (MINCLE) signaling pathway with GFP under control of nuclear factor kappa-B (NF-$\kappa$ B; left) and the dose dependence of the GFP expression of the cells (right). The input and output distribution of single-cell resolution data are divided by indexed (i.e. $i$ and $j$) rectangular grids to estimate the joint and marginal probability of the distribution. $N_{i,j}$ and $N_t$ are the number of cells in the index $(i,j)$ and the total number of cells, respectively. (**B**) Estimated joint probability distribution between the input glycan concentration and GFP expression using the procedures described in (**A**). The joint entropy ($H_{xy}$) and marginal entropies ($H_x$ and $H_y$) and therefore mutual information ($I_{xy}$) can be calculated from the estimated distribution. The channel capacity of the distribution can be further found by maximizing the mutual information with various trial input distributions (see also *Appendix 2—figure 3*). Experimental data for FurFurMan stimulation is shown. (**C–D**) Monoclonal reporter cells expressing dectin-1, dectin-2, MINCLE, or wild type (WT) were stimulated with (**C**) invertase or (**D**) FurFurMan (n=3). (**E**) Dose response of the dectin-2 reporter cells is shown both as geometric mean with SD and boxplot with the whiskers representing the 1 percentile of the cellular population (n=6).

cells. The channel capacity takes cellular variation and activation across a whole range of incoming stimulus of single-cell resolved data into account and quantifies all of that data into a single number.

Herein, we studied dectin-2, dectin-1, MINCLE, DC-SIGN, MCL, TNFαR (TNF alpha receptor), and TLR-1&2 in NF-κB reporter cells using single-cell resolved flow cytometry (*Figure 1A*, see also Appendix 2). To accurately quantify the information transmission in the receptors' signaling pathways in response to exogenous glycans, we use the channel capacity as a metric (*Figure 1B*). By employing channel capacity measurements, we found dectin-2 channel has relatively low signaling capacity, which in turn is synergistically increased in the presence of co-expressed MCL receptor. Furthermore, the channel capacity of dectin-1 and dectin-2 channel for the same glycan ligand was compromised when

both receptors are expressed by the cell while increasing the binding sensitivity ($EC_{50}$) to the ligand. Overall, our findings and approach provide a quantitative description of glycan lectin communication and signal integration of CTLs and other receptors, which may lead to a better understanding of key phenomena such as pathogen recognition and autoimmunity.

## Results

### Quantifying signal transduction in glycan-based communication

We employed a single-cell resolved reporter system to monitor CTL activity by GFP expression under control of the transcription factor NF-κB in human monocytic U937 cells (*Figure 1A*). Dectin-2 was expressed in these reporter cells, and stimulation was conducted using various ligands (*Figure 1C–E*). FurFurMan, an extract of *Malassezia furfur*, as well as the polysaccharide mannan and invertase, both from *Saccharomyces cerevisiae*, initiated dectin-2 signaling. In contrast, owing to the lack of multivalency, mannose itself could not initiate signaling but was able to inhibit dectin-2 function (*Figure 1C and D*, and also *Appendix 1—figure 1 A*; *Ishikawa et al., 2013*). In parallel, the invertase treated with α-mannosidase does not activate the NF-κB signaling, indicating the glycosylation-dependent dectin-2 activity (*Appendix 1—figure 1 B*). The activation of human dectin-2 receptor is in line with previous reports on its murine homolog, which is triggered by Man-α1–2 Man moieties presented on scaffolds like proteins, glycans, or polystyrene beads (*Ishikawa et al., 2013*; *Yonekawa et al., 2014*; *Zhou et al., 2018*). Analogously, introduction of dectin-1 into the reporter cells enabled detection of NF-κB-based GFP expression after stimulation. However, while FurFurMan could also stimulate dectin-1 cells, this was not inhibited by the addition of mannose, which is expected for this β-glucan receptor (*Figure 1D*).

Next, we studied the dose-response behavior of dectin-2 reporter cells stimulated with FurFurMan over a wide range of input concentrations (*Figure 1E*). The cellular population revealed an overlap between the unstimulated and the maximally stimulated population, demonstrating the absence of a clear two-state behavior on a population level (*Figure 1E*, *Appendix 1—figure 1 C*). To ensure that change in the reporter level is not affected by protein expression rate, we confirmed that GFP expression required at least 16 hr of stimulation to reach its maximum in steady-state protein expression, while short stimulation with for 2–6 hr does not lead the maximum level of GFP production (*Appendix 1—figure 1 D*). We also ruled out any influence of the selection process for the cellular clones, by sorting dectin-2 expressing cells according to their GFP expression level. When re-stimulated, both populations again showed the same broad GFP expression, confirming the wide range of the response to be independent of genetic differences between individual cells (*Appendix 1—figure 1 E*). Taken together, observing noisy dectin-2 signaling on a single-cell level in relevant model cell lines reveals a broad population distribution when stimulated.

### Dectin-2 transmits less information than other receptors

To investigate whether other receptors with similar signaling pathways follow the same principle, we analyzed the dose response of dectin-1, MINCLE, and the non-CTLs TNFαR and TLR-1 and -2 (*Bode et al., 2019*; *Holbrook et al., 2019*; *Ishikawa et al., 2009*; *Ozinsky et al., 2000*). To quantify the underlying signal transmission in a cellular population, the channel capacity was used as a metric. Note that we choose the stimulation time, the period of incubation time of the cell with the input ligands, as the time point when GFP response and channel capacity reach the maximum and steady-state value (*Appendix 1—figure 2 A and B*). And therefore, the stimulation was 13 and 16 hr for TNF-α and the rest of the ligands, respectively. Previous work on TNF-α signaling found the TNF-α channel to have a channel capacity of about 1 bit in particular 1.64 bits when a reporter cell system was used (*Cheong et al., 2011*). In addition, this channel capacity can be further increased if one can measure the temporal evolution of output dynamics instead of static output dataset (*Selimkhanov et al., 2014*). Such channel capacity suggests that a cellular population can use a receptor to distinguish between two states: on/off or presence/absence of a stimulant. For U937 cells, we found the TNFαR transmits 1.34 bits of channel capacity for TNF-α stimulant (*Figure 2A and B*), which was not influenced by the introduction of additional lectins (i.e. MINCLE, dectin-2, and DC-SIGN, see *Appendix 1—figure 2 C*). In the case of dectin-1 expressing U937 cells, the channel capacities were 1.20 and 1.09 bits for depleted zymosan (DZ) and FurFurMan input, respectively, while both MINCLE

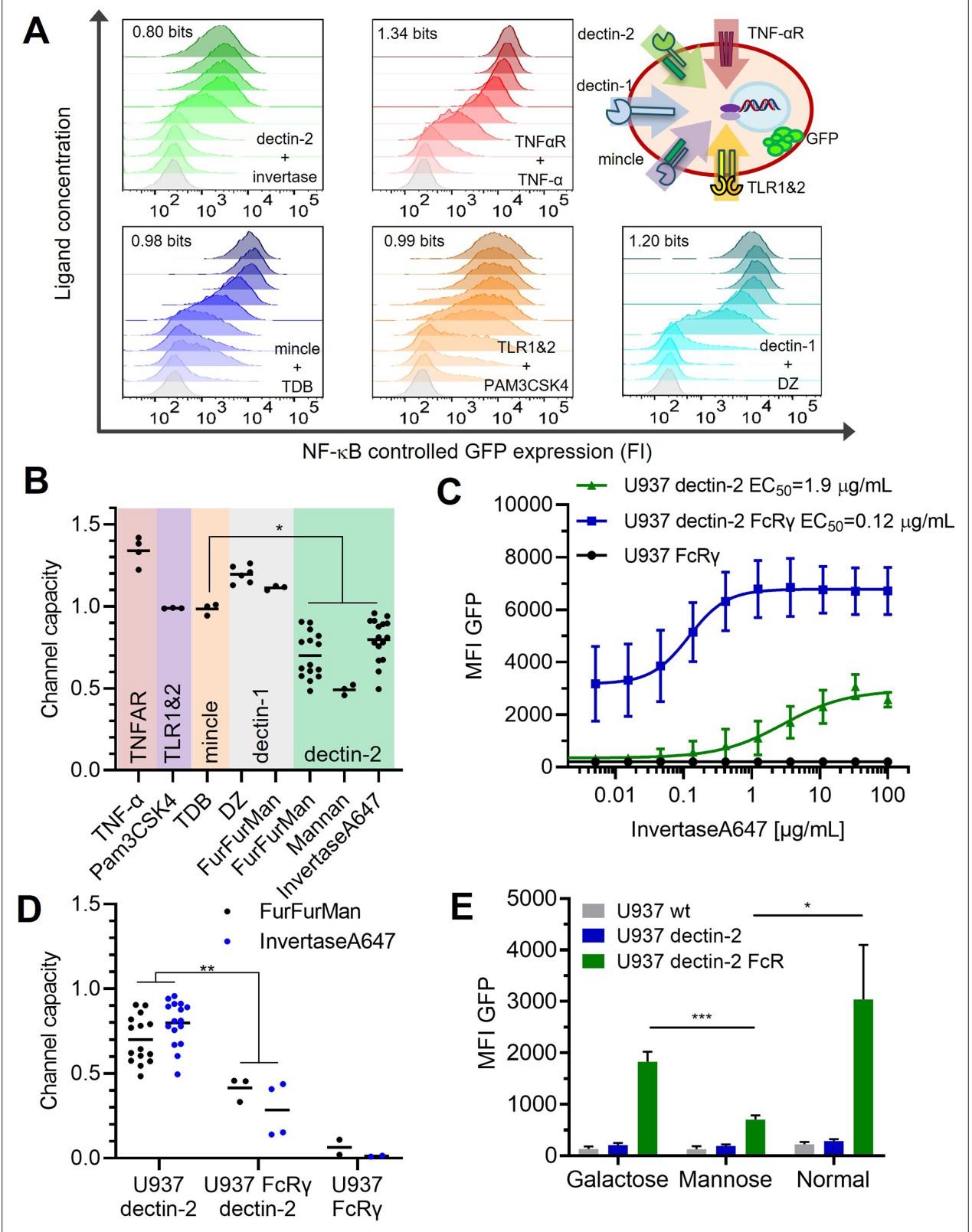

**Figure 2.** Quantification of signal transduction. (**A**) Representative histograms of U937 reporter cells dose response, stimulated specifically with invertase for dectin-2, TNFα for the TNFαR, trehalose-6,6-dibehenate (TDB) for macrophage-inducible C-type lectin (MINCLE), Pam3-Cys-Ser-Lys4 (Pam3CSK4) for TLR1&2, and depleted zymosan (DZ) for dectin-1. The number in each histogram is the channel capacity of the corresponding signals. Top right panel shows a schematic representation of the five analyzed receptor channels. (**B**) Estimated channel capacities between various pairs of

*Figure 2 continued on next page*

Figure 2 continued

ligand and receptor (*p<0.05, Wilcoxon rank-sum test). (**C**) Monoclonal reporter cells either expressing dectin-2 (n=3), FcRγ (n=2), or dectin-2 and FcRγ (n=4) were stimulated for 16 hr with various concentrations of invertase, labeled with Atto647 dye. The error bars indicate the SDs. (**D**) Channel capacities from stimulation with invertase and FurFurMan (data also seen in B) and FurFurMan stimulation of U937 reporter cells (**p<0.01, Wilcoxon rank-sum test). (**E**) Unstimulated reporter cells (mock stimulated) 16 hr after cultivation with 25 mM galactose, or mannose, or under normal conditions for 48 hr (n=3, *p<0.05 and ***p<0.001, Student's *t*-test), suggesting that dectin-2 mediated self-recognition leads to a high basal level of cellular activation in FcRγ overexpression cells.

and TLR1&2 had a channel capacity of 0.98 and 0.99 bits, respectively. Since these receptors signal via NF-κB, these differences can be explained by receptor expression levels and downstream pathways. In contrast, dectin-2 stimulation resulted in a channel capacity of 0.70 bits using FurFurMan as a ligand. Stimulation using heat inactivated invertase or mannan had 0.80 and 0.49 bits, respectively (**Figure 2B**). Also, in THP-1 cells, a similar trend of lower GFP expression upon stimulation is observed, further supporting the notion that dectin-2 has a lower signal transmission capacity compared to the other receptors such as TNFαR (**Appendix 1—figure 2 D-F**).

The most striking difference was found between MINCLE and dectin-2, as both lectins use the same signaling pathway via FcRγ (**Ishikawa et al., 2013**), suggesting that the substantial differences between the channel capacities rely on very early ligand recognition events. We hypothesized overexpression of the signaling protein FcRγ might increase the information transmitted via dectin-2. The overexpression of FcRγ resulted in at least twofold increase of NF-κB controlled GFP expression (**Figure 2C**). Overexpression of both dectin-2 and FcRγ yielded a high-basal NF-κB activation of the cells while the sensitivity for its ligand (EC$_{50}$) increased about 50-fold. While the maximal GFP signal of dectin-2 (MFI, mean fluorescence intensity) was increased in the presence of FcRγ overexpression, the channel capacity however decreased simultaneously (0.41 bits; **Figure 2 C–E**). Since competition with mannose reduced this effect, we speculate that decreased channel capacity might originate from self-recognition of dectin-2 of ligands being present either on the same cell or those in close proximity during the culture conditions (**Figure 2E**). From this, we concluded the channel capacity of a glycan-based communication channel is not necessarily coupled to its sensitivity. Also, the ability of a communication channel to transmit information is not well described by its maximal signal alone (i.e. MFI), but rather by the channel capacity. Next, we quantified the number of receptors and excluded that the difference in MINCLE and dectin-2 channel capacities is due to differences in receptor expression levels (**Appendix 1—figure 2 G**). Taken together, dectin-2 has relatively less channel capacity, and while its sensitivity (EC$_{50}$) can be modulated with FcRγ, the transmitted information does not increase. Additionally, the number of receptors has little influence on the channel capacity or amplitude.

## Signal integration compromises between dectin-1 and dectin-2 receptors when both are engaged

To expand our insight from isolated cell surface receptors to the interplay between multiple lectins, we prepared reporter cells expressing dectin-2 and dectin-1 simultaneously. FurFurMan served as a stimulant since it interacts with both dectin-1 and dectin-2. First of all, we found that the level of receptor expression did not change upon expression of an additional lectin (**Figure 3A**). Dectin-1 expressing cells gave a higher maximal signal (i.e. maximal MFI) and channel capacity than dectin-2 expressing cells; however, the latter channel showed higher sensitivity (EC$_{50}$) to FurFurMan. We found that the double positive cells did compromise between the two receptors, displaying the values corresponding to the intermediate values of the EC$_{50}$ and channel capacity of dectin-1 and dectin-2 (**Figure 3B and C**). Additionally, mannose could be used to interfere with dectin-2 signaling, thus U937 dectin-1 dectin-2 expressing cells showed the same dose-response curve as dectin-1 expressing cells (**Figure 3D**). When DZ, a dectin-1 specific stimulants, was used, dectin-2 expression did not significantly influence the response of the double positive cells. Hence, dectin-2 specific signaling was not influenced by dectin-1 expression (**Appendix 1—figure 3 A-C**). Moreover, inhibition of dectin-2 signaling initiated by FurFurMan by the addition of 25 mM mannose resulted a response that was not a compromise. Taken together, we see that the simultaneous stimulation of dectin-1 and dectin-2 resulted in a compromise between their channels, which demonstrates how these two channels integrate glycan signal into response.

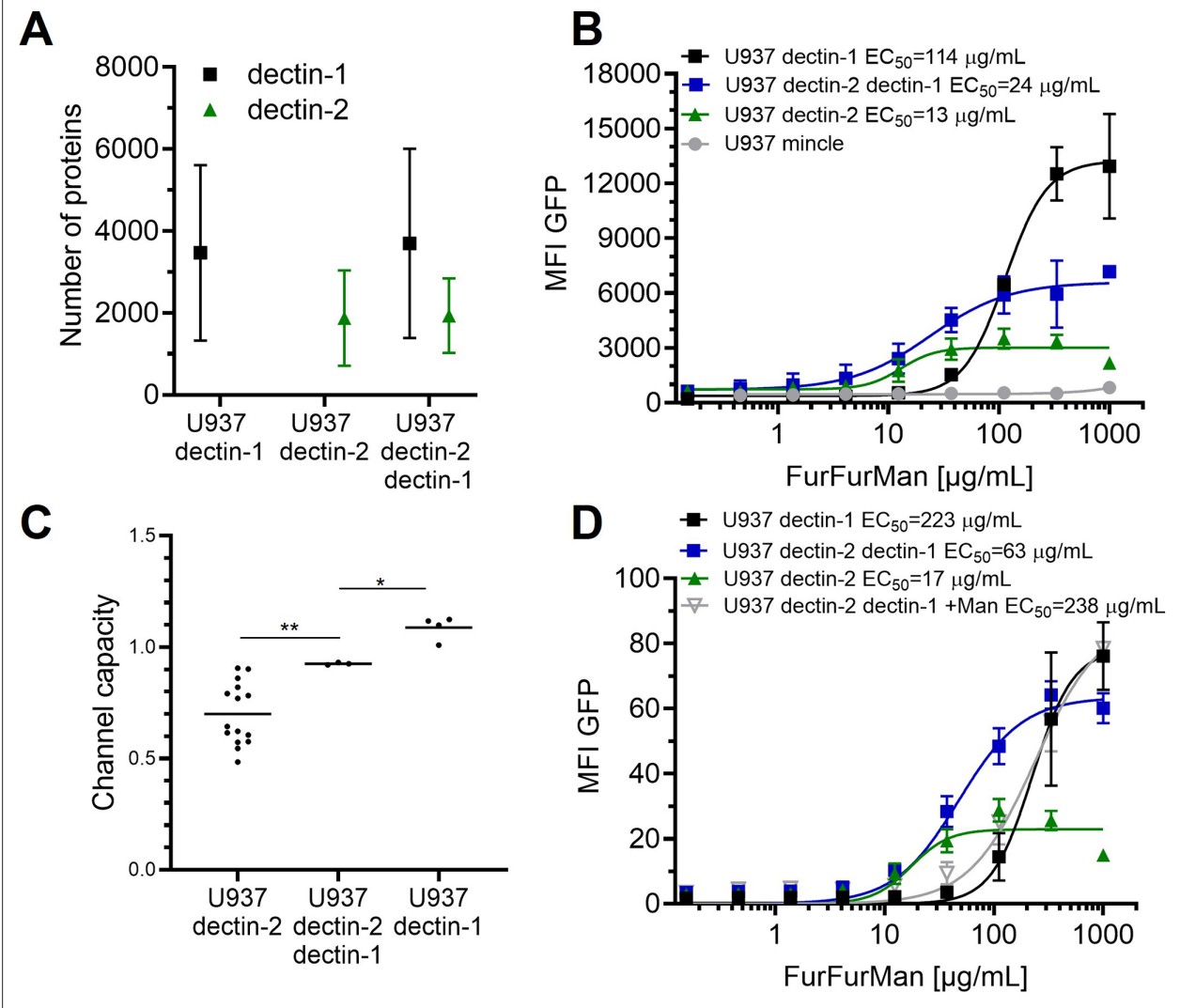

**Figure 3.** Signal integration of dectin-1 and dectin-2. (**A**) Quantitation of surface expression of U937 dectin-1, dectin-2, and dectin-1 dectin-2 U937 reporter cells. Fluorescence intensity (FI) values were transformed into the number of proteins expressed using a PE-quantitation. Graph shows geometric mean ± robust SD of the cellular population. (**B**) Monoclonal reporter cells either expressing macrophage-inducible C-type lectin (MINCLE), dectin-2, dectin-1, or both dectin-2 and dectin-1 were stimulated for 16 hr with various concentrations of FurFurMan (n≥3). The error bars indicate the SDs. (**C**) Channel capacities of U937 reporter cells expressing either dectin-1, dectin-2, or both stimulated with FurFurMan (*p<0.05 and **p<0.01, Wilcoxon rank-sum test). (**D**) Monoclonal reporter cells either expressing dectin-2, dectin-1, or both dectin-2 and dectin-1, stimulated with various concentrations of FurFurMan (n=3). Dectin-1 expressing cells were stimulated either with or without 25 mM of mannose. The error bars indicate the SDs.

## Macrophage C-type lectin (MCL) increases the channel capacity of dectin-2

To further expand our insights into signal transmission through multiple lectins, we wondered whether co-expression of other lectins would synergistically increase the channel capacity of dectin-2 signaling. For this, we included DC-SIGN and MCL (*Figure 4A*). Although DC-SIGN does not elicit NF-κB signaling by itself in U937 cells, it is known to recognize high-mannose structures present on invertase (*Gringhuis et al., 2009*). As expected, U937 dectin-2 DC-SIGN cells experience significantly increased ligand binding (*Appendix 1—figure 4 A and B*). We then speculated that this would either (a) promote the ligand recognition by pre-concentration of the stimulants on the cell surface or (b) sequester the input signal from dectin-2, reducing the cellular response. In fact, DC-SIGN-mediated ligand binding did not alter the dectin-2 channel capacity for FurFurMan or invertase stimulation or did DC-SIGN expression itself modulate TLR4 signaling (*Figure 4B*, *Appendix 1—figure 4 C*). However, the sensitivity, as assessed by EC50, increased for dectin-2 DC-SIGN expressing cells (*Figure 4A*).

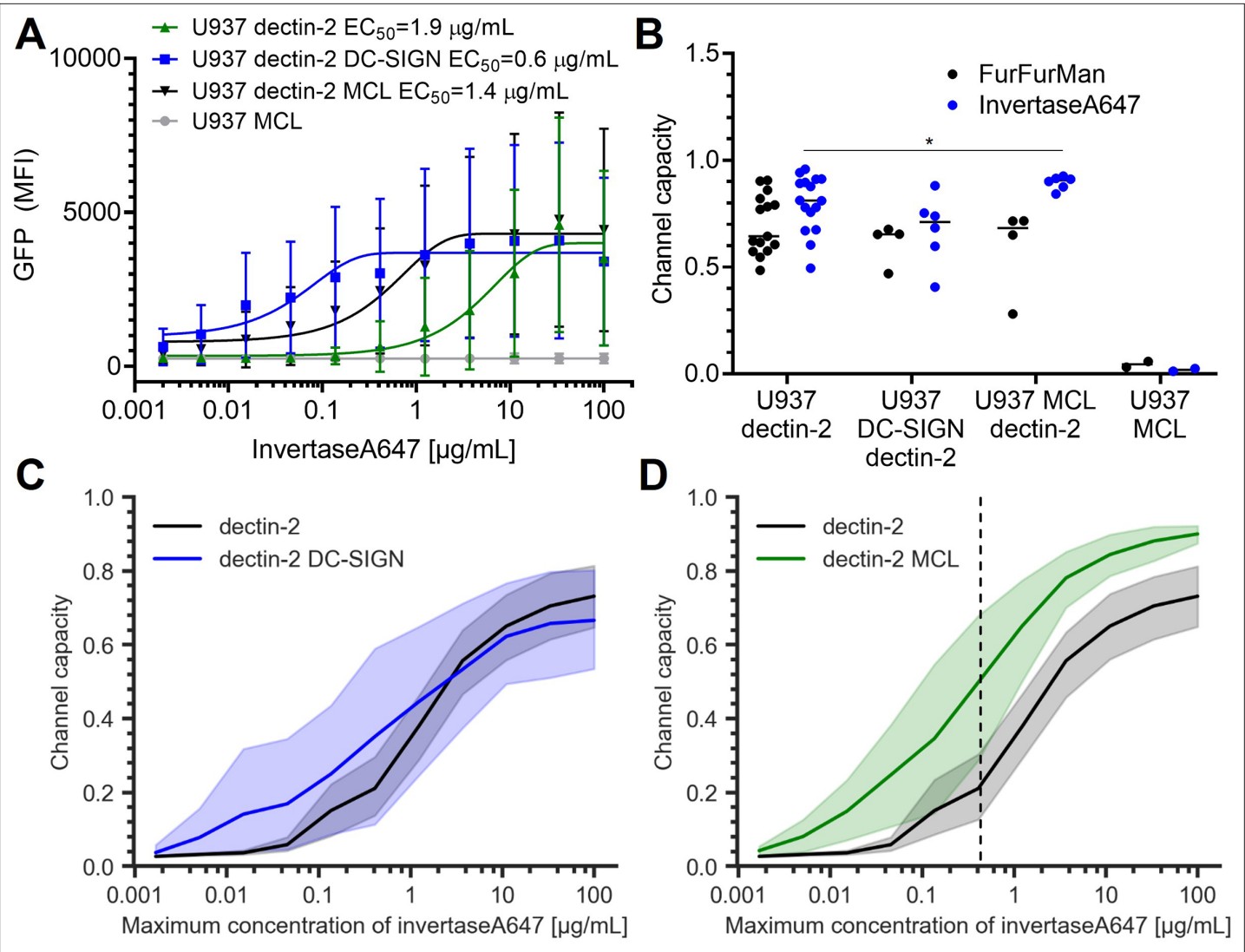

**Figure 4.** Signal response of dectin-2 in the presence DC-specific ICAM-3–grabbing nonintegrin (DC-SIGN) or macrophage C-type lectin (MCL). (**A**) Representative dose response of invertase stimulation of U937 cells expressing dectin-2, MCL, or dectin-2 co-expressed with either DC-SIGN or MCL. (**B**) Channel capacities of dectin-2 in combination with DC-SIGN and MCL after stimulation with either FurFurMan or invertase (*p<0.05, Wilcoxon rank-sum test). (**C** and **D**) Channel capacities calculated from different maximum invertase concentrations of dectin-2 expressing cells compared with either DC-SIGN (**C**) or MCL (**D**) co-expression. The shaded regions represent the 95% CI of the channel capacity. The right side of the dashed line in (**D**) is the region that shows statistical significance between dectin-2 and MCL co-expressed dectin-2 (*p<0.05, Wilcoxon rank-sum test).

The increased sensitivity due to DC-SIGN co-expression might increase the channel capacity if the allowed dose range spans low-concentration region. Therefore, we calculated the channel capacity by increasing the maximum input concentration. However, this was not the case (*Figure 4C*). Contrary to DC-SIGN, overexpression of MCL significantly increased the channel capacity of dectin-2 expressing cells, particularly when limiting our dataset to low-maximum invertase concentrations (*Figure 4B and D*, *Appendix 1—figure 4 D and E*). This indicates that MCL enhances the fidelity of invertase information transmission of dectin-2 channel, providing quantitative measurement of synergistic effect of MCL (*Ostrop et al., 2015*; *Zhu et al., 2013*).

We then wondered whether the difference in channel capacity between dectin-2 and TNFαR could simply be a result of affinity. Since TNFαR has a nanomolar affinity for its ligand (*Grell et al., 1998*), we applied an anti-dectin-2 antibody to stimulate dectin-2 cells. Even under these conditions, we did not monitor an increase in channel capacity (*Appendix 1—figure 4 F*). Therefore, we found that MCL

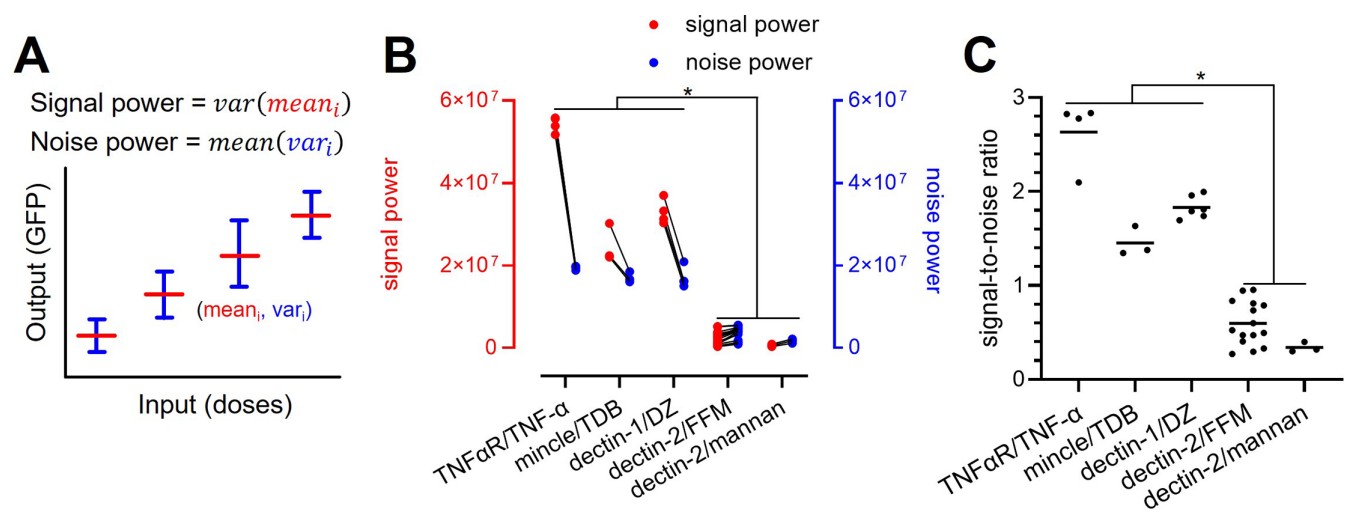

**Figure 5.** Decomposition of the signaling channels into signal power and noise power. (**A**) Schematic description of signal and noise power. mean_i and var_i are the average and variance of the output at *i*th dose, respectively. (**B**) Decomposed signal power (red) and noise power (blue) of the individual signaling channel. (**C**) The ratio between signal power and noise power (i.e. signal-to-noise ratio) of individual channel given in (**A**) (*p<0.05, Wilcoxon rank-sum test).

but not DC-SIGN significantly increase the dectin-2 channel capacity, while both MCL and DC-SIGN enhance cellular binding of the stimulants and the resulting cellular sensitivity to invertase.

## Dectin-2 channel has a low signal-to-noise ratio

The relatively low channel capacity of dectin-2 could be a result of its limited maximum GFP expression even at high-stimulant concentrations compared to the other channels (*Figure 2B*). For this, we define the signal power as the variation of the mean GFP expression under individual stimulant dose (*Figure 5A*). In addition, the level of background noise (i.e. noise power) of the channel can be defined as the average of the variance of GFP expression at a given stimulant dose. These definitions allow to decompose signal and noise power (Appendix 3) and analyze them separately to infer how those two parameters shape the channel capacity.

TNFaR, MINCLE, and dectin-1 have a similar level of noise power. Amongst the three receptors, TNFaR shows the highest signal power and consequently the highest signal-to-noise ratio (*Figure 5B and C*). All three channels have a signal-to-noise ratio higher than one. For dectin-2, both signal and noise power are low compared to the other receptors; however, the noise power exceeds the signal power, resulting in a significantly lower signal-to-noise ratio. Since the signal power is independent of the noise power, our data indicate that the lower variation of the mean GFP expression (i.e. signal power) of dectin-2 dictates the reduced channel capacity compared to the other receptors. A similar conclusion cannot be drawn for the noise power since it is inherently coupled to the signal power (see Appendix 3).

We further employed the decomposition method to dectin-2 signaling in the presence of either dectin-1 co-expression or FcRγ overexpression (*Appendix 1—figure 5*). Analogous to the compromised channel capacity when dectin-1 and dectin-2 were co-expressed (*Figure 3C*), the analysis revealed that both the signal and the noise power were compromised as well (*Appendix 1—figure 5 A and B*). In case of FcRγ overexpression, dectin-2 signaling after invertase stimulation is characterized by increased noise power, resulting in decreased signal-to-noise ratio (*Appendix 1—figure 5 C-E*). Therefore, despite the high-GFP expression at high-stimulant concentrations (*Figure 2C*), the overexpression of FcRγ, as additional signaling hubs, involved in the dectin-2/NF-κB cascade did not increase the signal power but instead elevated the noise power, leading to reduced channel capacity. Taken together, the relatively low-channel capacity of dectin-2 is directly related to its low-signal power, and the overexpression of FcRγ further decreases the channel capacity through increasing the noise power.

## Discussion

We set out to better understand how glycan-encoded information is read in cellular communication. We established a glycan-responsive in vitro model and exploited the channel capacity as a quantitative metric. For the receptors other than dectin-2, the channel capacities were around 1 bit or higher, similar values that have been reported for other systems previously (*Suderman et al., 2017*). In particular, TNF-α receptor has a channel capacity of 1.64±0.36 bits, which was found in a comparable reporter cell system (*Cheong et al., 2011*). Interestingly, the number of receptors expressed on the cell surface did not determine the channel capacity of a signaling channel (*Appendix 1—figure 2 G*). Our results exemplify that lectin signaling pathways and especially the dectin-2 pathway should not be viewed as a deterministic on/off-switch, but rather as difference in the probability of cells to be active at a certain dose. This is in line with previous reports strengthening a quantitative view of cellular signaling and taking the cellular microheterogeneity into account (*Levchenko and Nemenman, 2014*; *Zhang et al., 2017*). We found that the mannose binding CTL dectin-2 to transmits less information compared to other receptors of the same family (*Figure 2B*).

To understand how these insights could be expanded on the interplay between multiple receptors like the CTLs occur on innate immune cells rather than isolated lectins, we employed combinations of CTLs on our model cells. Dectin-2 and dectin-1 recognize different epitopes on FurFurMan, and we found that the effects were not additive, but a compromise between the two receptors, showing intermediate sensitivity (EC$_{50}$) and channel capacity between dectin-2 and dectin-1 (*Figure 3A and B*). This effect implies at high concentrations of FurFurMan the dectin-2 channel is actively inhibiting dectin-1 signaling, resulting in a lower cellular NF-κB activation. It is well known that lectins are able to modulate the signals of other receptors (*Geijtenbeek and Gringhuis, 2009*; *Gringhuis et al., 2009*; *Miyake et al., 2015*). Yet this compromise is an exciting discovery since to the best of our knowledge previous studies have not quantified lectin signal integration. Hence, it is likely that during a fungal infection, exposing multiple epitopes of pathogens are recognized by the precise arsenal of immune receptors, and their underlying signaling pathways are integrating the information contained within the epitopes. This in turn leads to a compromise of all activated receptors and results in a specifically tailored biochemical response of the given immune cell (*Ostrop and Lang, 2017*).

Dectin-2 itself we found to have relatively less channel capacity when compared to the closely related MINCLE that uses the same pathway with more signal power (*Figure 2A and B*). It is therefore likely the receptor itself determines very early on the information flow into the cell. This could be a result of MINCLE being stimulated with crystalline insoluble ligands which could result in larger signaling clusters at the cellular surface. Alternatively, dectin-2 signaling could be influenced by mannose structures that are present on the cellular surface by giving rise to background signaling and selection for reducing signaling power in an in vitro setting of high-cellular density. Additionally, since dectin-2 binds high-mannose structures of eukaryotic origin (*McGreal et al., 2006*), a too sensitive reaction might lead to permanent self-recognition of human Man9 structures for example and hence potential autoimmune reactions. This hypothesis is supported by the dectin-2-dependent high-basal activity of FcRγ overexpressing dectin-2 cells, which in turn is responsible for a lower channel capacity in dectin-2 FcRγ cells (*Figure 2C–E*). Hence, dectin-2 could have evolved to use the CARD9-BCL-10-Malt1 pathway to NF-κB less effective. Along the same lines, recent reports show that CTLs are in general becoming more important in autoimmunity, dectin-2 in particular is known to be responsible for the development of allergic reactions (*Dambuza and Brown, 2015*; *Parsons et al., 2014*).

We first thought a combination of multiple lectins might synergistically enhance signaling capacity of dectin-2. But while DC-SIGN greatly enhanced ligand binding to the cells, meaning the increased sensitivity (EC$_{50}$), it did not significantly increase the channel capacity (*Figure 4A–C*, *Appendix 1—figure 4A and B*). In contrast to DC-SIGN, the closely related MCL to dectin-2 has a significant synergetic effect on dectin-2 channel capacity at particularly low-stimulant concentrations, potentially making double positive cells more discriminative, at earlier timepoints of infection compared to dectin-2 expressing cells, substantiating the importance of signal integration to understand an cellular innate immune response (*Ostrop and Lang, 2017*).

Finally, it is important to take into consideration that our conclusions came from model cell lines, which were used as a surrogate for cell-type-specific lectin expression patterns of primary immune cells. Human monocytes and dectin-2 positive U937 cells have comparable receptor densities and respond similar to stimulation with zymosan particles (*Appendix 1—figure 6A and B*). Importantly,

since our channel capacity calculations are applicable regardless of the nature of signal and medium, one could use it to quantify cellular responses in similar assays in the future. Work is ongoing to address central questions of cellular communication based on glycan lectin interactions.

## Materials and methods

All reagents were bought from Sigma Aldrich, if not stated otherwise.

### Reporter cell generation and reporter cell assay

U937 cells were transduced with an NF-κB-GFP Cignal lentivirus (Qiagen) according to the manufacturer's instructions to generate NF-κB reporter cells. 0.5 mL of 2e5 cells were mixed with the lentivirus at an MOI (multiplicity of infection) of 15 and spin transduced for 1.5 hr at 33°C and 900 g. After 48 hr of rest, cells were selected with puromycin (gibco) for three passages. Eight cultures from a single cell each were subsequently made and evaluated according to their GFP expression, clone #5 only monoclonal cells were used for all experiments of this paper.

### Reporter cell assay

U937 reporter cells were used in its log phase, and 100 μL were plated in a 96-well plate with 3e4 cells per well. Cells were challenged in complete media (RPMI with 10% FBS (fetal bovine serum), 1% Glutamax, 1% Pen/Strep, and all by gibco) with TNF-α and various other ligands and at various concentrations for 13 hr and 16 hr, respectively. After incubation, cells were re-suspended once in DPBS (Dulbecco's phosphate-buffered saline) and the expressed GFPs fluorescent intensity was measured by flow cytometry (Attune Nxt, Thermo Fisher).

### Cell culturing and passage

U937 cells were kept between 1e5 and 1.5e6 cells/mL in complete media with passage 2–4 times a week. 293 F cells were adherently cultured in DMEM (Dulbecco's modified Eagle's medium) with 10% FBS, 1% Glutamax, 1% Pen/Strep (Gibco), and split 2–3 times per week. All cells were tested for mycoplasma contamination using Minerva biolabs VenorGeM Classic.

### Antibody staining and quantitation

For the surface staining, cells were incubated in with the respective antibodies and isotype controls for 30 min at 4°C in DPBS, then washed once in DPBS +0.5% BSA and measured via flow cytometry. For perforated stains cells were first fixed in 4% PFA (Carl Roth) at 4°C for 20 min, then perforated in perforation solution (DPBS +0.5% BSA+0.1% Saponine) for 20 min at 4°C. The cells were then re-suspended in perforation solution containing the respective antibodies, incubated for 20 min at 4°C and measured via flow cytometry after being washed once. To quantify the fluorescent intensities, we used the BD PE quantitation kit, which allowed us to calibrate FI to the number of PE molecules present in a sample. A list of all used antibodies can be found in the *Supplementary file 1*.

### Generation of lectin overexpressing cells

cDNA of MINCLE, dectin-2, MCL, FcRγ, dectin-1, and DC-SIGN were cloned into vector BIC-PGK-Zeo-T2a-mAmetrine:EF1a as previously reported (*Wamhoff et al., 2019*). This bicistronic vector expresses mAmetrine under the PGK promoter. To combine multiple GOI (gene of interest), we also used the lentiviral vector EF1a-Hygro/Neo a gift from Tobias Meyer (Addgene plasmid # 85134). Briefly, 293F cells were transfected with vectors coding for the lentivirus and GOI. Lentivirions were generated for 72 hr, and the supernatant was frozen to kill any remaining 293 F cells. This supernatant was used to transduce the GOI into U937 cells via spin infection at 900 g and 33°C in the presence of 0.8 μg/mL polybrene (*van de Weijer et al., 2014*). After 48 hr of rest, the U937 cells were selected with appropriate antibiotics (Zeocin 200 μg/mL, G418 500 μg/mL, or Hygromycin B 200 μg/mL; Thermo Fisher, Carl Roth, Thermo Fisher, respectively). A list of used primers can be found in the *Supplementary file 2*.

### Labeling of proteins

Invertase (5 mg in 1 mL) was heat inactivated for 40 min at 80°C and mixed with 3×molar excess of Atto647N-NHS dye (AttoTech) according to the manufacturer's protocol. The labeled protein was

purified using Sephadex G-25 column, and aliquots were frozen at –80°C. Since we found the labeled invertase to contain less impurities, we used Atto647 labeled invertase for all experiments shown in this study. Human TNF-α (Peprotech) was labeled with the same procedure, yet without heat inactivation. The degree of labeling was determined to be around 1 as determined with a labeled protein concentration measurement of a NanoPhotometer NP80 (Implen).

### Channel capacity calculation

Calculations of channel capacity were based on *Cheong et al., 2011* and *Suderman et al., 2017*. See Appendix 2 on channel capacity calculation for details.

### Data representation, software, and statistical analysis

Data is shown as mean ± SD. Statistical analysis of data was performed by unpaired two-tailed t-test, with significant different defined as ($p<0.05$). $EC_{50}$ values were calculated in graph pad prism version 8.4.2 using four parametric dose vs. response function. When necessary statistical differences between $EC_{50}$ values were compared using an extra-sum-of-squares F test. Detail of statistical tests and $EC_{50}$ determinations can be found in the SI raw data file. FlowJo v.10 was used for analysis and export of flow cytometry data.

### Data availability

All data is available at Dryad. The Jupyter notebook including the channel capacity calculation and noise analysis is available at: https://github.com/imaginationdykim/2022.CC, (copy archived at *Fuchsberger, 2023*).

## Acknowledgements

This project (GLYCONOISE) has received funding from the European Research Council (ERC) under the European Union's Horizon 2020 research and innovation program (Grant agreement No. 716024). We thank Max Planck Society for support and Prof. Dr. Peter H Seeberger for helpful discussions. We also thank the Deutsches Rheuma Forschungszentrum (DRFZ) for providing access to their cell sorting facility. The computational results presented were obtained using the CLIP cluster (https://clip.science).

## Additional information

### Funding

| Funder | Grant reference number | Author |
| --- | --- | --- |
| European Research Council | 716024 | Christoph Rademacher |

The funders had no role in study design, data collection and interpretation, or the decision to submit the work for publication. Open access funding provided by Max Planck Society.

### Author contributions

Felix F Fuchsberger, Conceptualization, Formal analysis, Validation, Investigation, Visualization, Writing – original draft, Writing – review and editing; Dongyoon Kim, Conceptualization, Resources, Data curation, Software, Formal analysis, Investigation, Visualization, Methodology, Writing – original draft, Writing – review and editing; Natalia Baranova, Conceptualization, Formal analysis, Investigation, Writing – review and editing; Hanka Vrban, Investigation; Marten Kagelmacher, Validation, Investigation; Robert Wawrzinek, Formal analysis, Investigation, Writing – review and editing; Christoph Rademacher, Conceptualization, Supervision, Funding acquisition, Investigation, Writing – original draft, Project administration, Writing – review and editing

### Author ORCIDs

Felix F Fuchsberger ⬡ http://orcid.org/0000-0002-9379-9792

Dongyoon Kim  http://orcid.org/0000-0001-9162-8730
Natalia Baranova  http://orcid.org/0000-0002-3086-9124
Christoph Rademacher  http://orcid.org/0000-0001-7082-7239

**Decision letter and Author response**
Decision letter https://doi.org/10.7554/eLife.69415.sa1
Author response https://doi.org/10.7554/eLife.69415.sa2

## Additional files

### Supplementary files
• Supplementary file 1. List of antibodies.
• Supplementary file 2. List of primers.
• Transparent reporting form

### Data availability
We have uploaded the raw data of the study to Dryad at https://doi.org/10.5061/dryad.18931zd2g. Our scripts for data evaluation are also linked to GitHub and stated in the manuscript.

The following dataset was generated:

| Author(s) | Year | Dataset title | Dataset URL | Database and Identifier |
|---|---|---|---|---|
| Rademacher C | 2023 | Data from: Information transfer in mammalian glycan-based communication | https://doi.org/10.5061/dryad.18931zd2g | Dryad Digital Repository, 10.5061/dryad.18931zd2g |

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

## Appendix 1

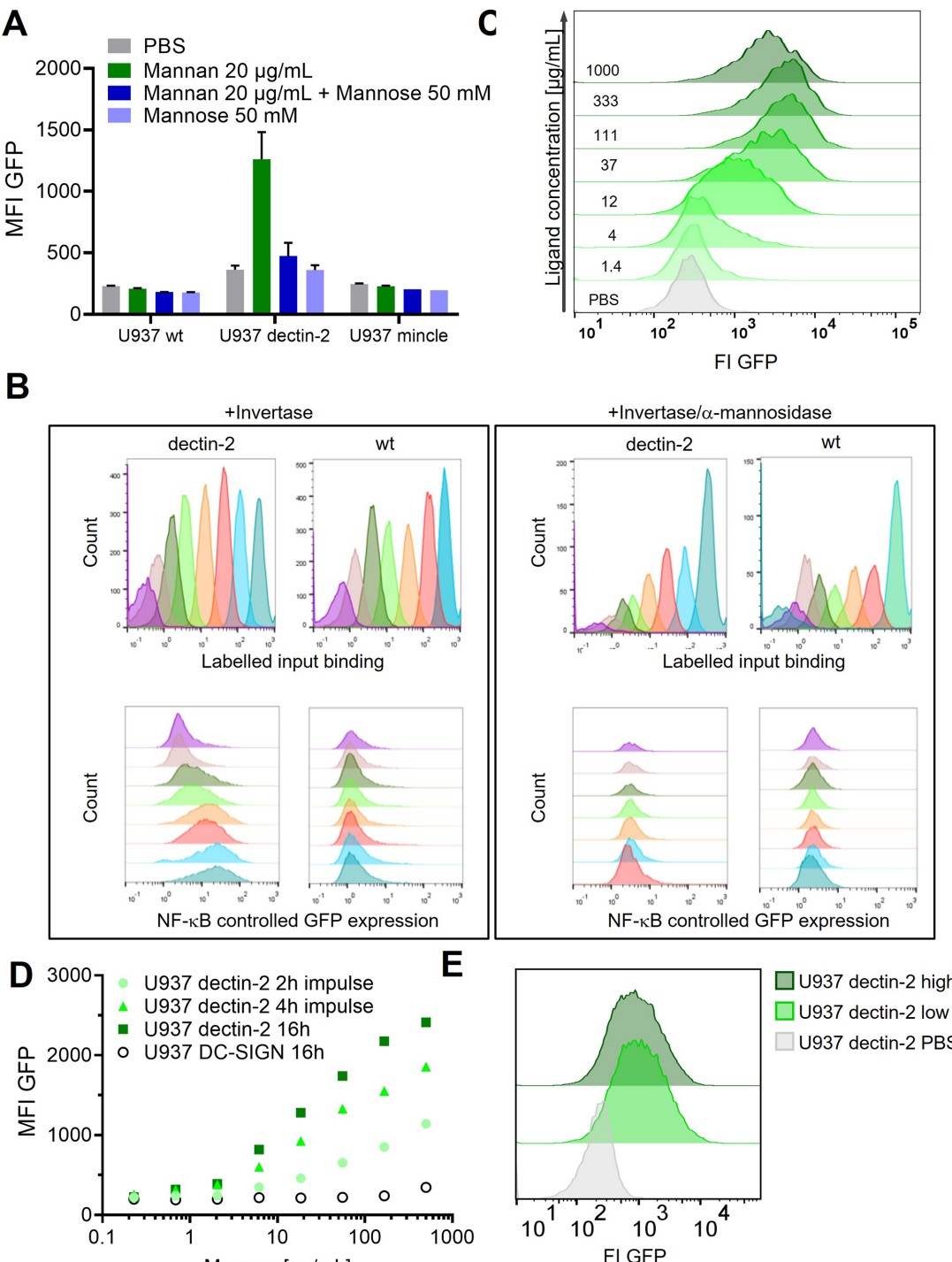

**Appendix 1—figure 1.** Signal transduction in the reporter cells. (**A**) Monoclonal U937 reporter cells expressing macrophage-inducible C-type lectin (mincle), dectin-2, or wild type were stimulated with mannan (n=4). Mannose alone could not stimulate dectin-2 but could inhibit stimulation by Mannan. (**B**) Dectin-2 downstream signaling is mediated by glycosylation of invertase. (Left) Interaction of Atto647 labeled invertase with dectin-2 or wild type U937 cells. While activation of the downstream signaling is dectin-2 specific (lower panel), the binding of invertase to U937 does not depend only on the interaction with dectin-2 (upper panel). Right: The invertase and U937 cells were treated with α-mannosidase. Such treatment resulted in complete inhibition of nuclear factor kappa-B
*Appendix 1—figure 1 continued on next page*

*Appendix 1—figure 1 continued*

(NF-κB) activation in dectin-2 cells (lower panel) but did not affect protein binding (upper panel). (**C**) Histograms of the dose response in *Figure 1D* and U937 dectin-2 expressing reporter cells react to various concentrations of FurFurMan. Darker histograms were stimulated with higher ligand concentration. (**D**) Dose response of dectin-2 and DC-specific ICAM-3–grabbing nonintegrin (DC-SIGN) expressing reporter cells stimulated for 16 hr, or stimulated for 2 and 4 hr, washed in fresh media, and incubated to a total of 16 hr. (**E**) U937 dectin-2 reporter cells were sorted in a GFP high and low population after stimulation for 16 hr with 300 µg/mL Mannan. The sorted cells were the re-stimulated 2 weeks later with 500 µg/mL.

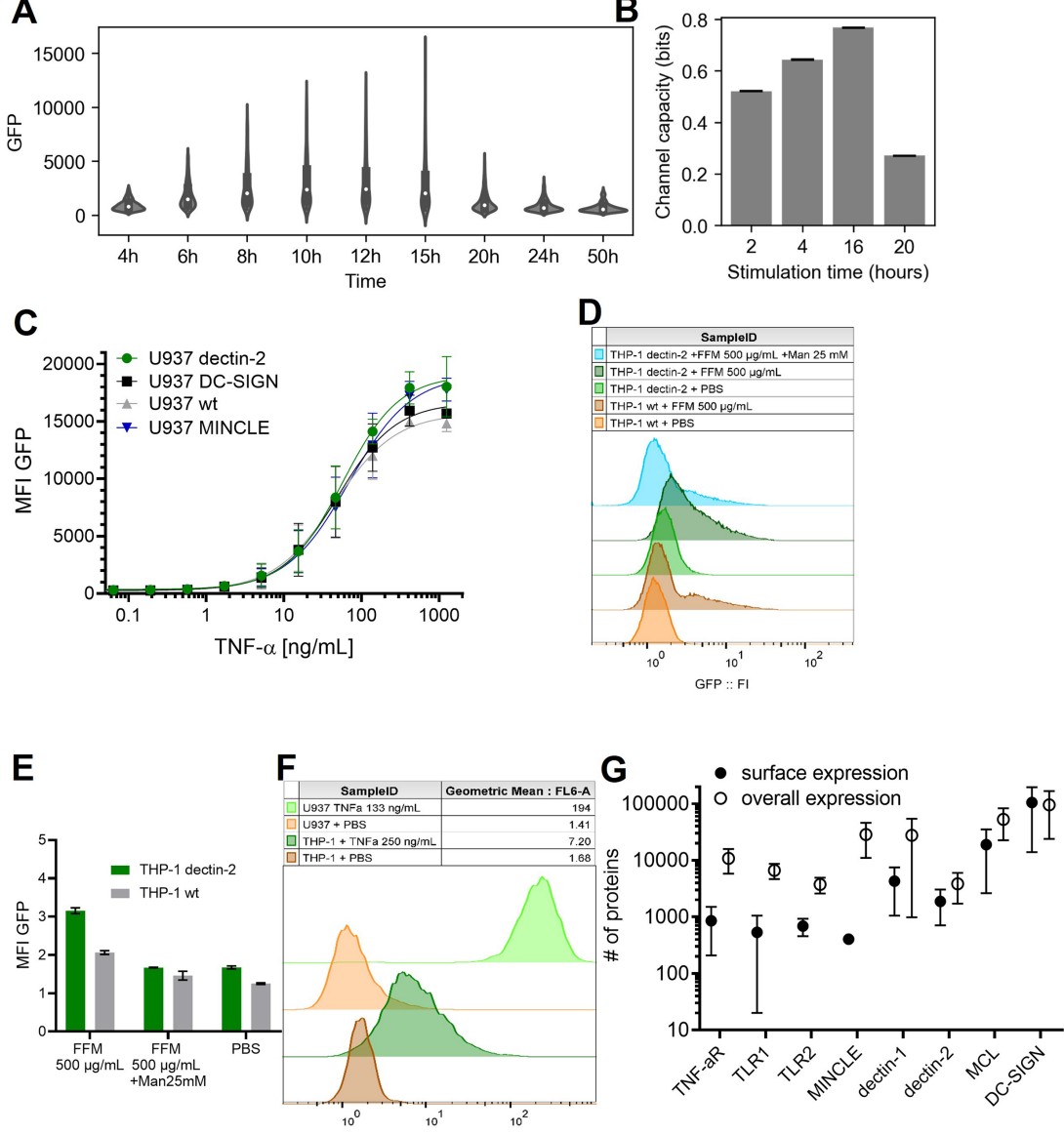

**Appendix 1—figure 2.** Stimulation time and stimulant dependent signal transduction in the model cells expressing the receptor of interest. (**A**) Violin plot of nuclear factor kappa-B (NF-κB) controlled GFP expression of U937 cells in response to 50 ng/mL TNF-α stimulant for various stimulation time. (**B**) Channel capacity of TNFαR channel to TNF-α stimulant for various stimulation time. Error bars indicate 95% CI. (**C**) Reporter cell lines expressing various lectins stimulated with TNF-α. (**D**) THP-1 reporter cells expressing dectin-2 or wild type (WT) were stimulated for 48 hr with FurFurMan (FFM), unstimulated (PBS), or the FurFurMan stimulation was inhibited with 25 mM mannose. Graph shown representative histograms. (**E**) Geometric means of the experiment done in A in triplicates (n=3) with the error bar representing SD. (**F**) Representative histograms showing the TNF-α stimulation (13 hr) of U937 and THP-1 reporter cells. THP-1 cells stimulated for 48 hr with TNF-α gave less signal than at 13 hr

*Appendix 1—figure 2 continued*
(data not shown). (**G**) Quantitation of surface and overall expression of receptors used in this study in U937 reporter cells. Cells were stained either for their surface expression or their overall protein expression with PE coupled antibodies. FI values were transformed into the number of proteins expressed using a PE-quantitation kit. Graph shows geometric mean ± robust SD of the cellular population.

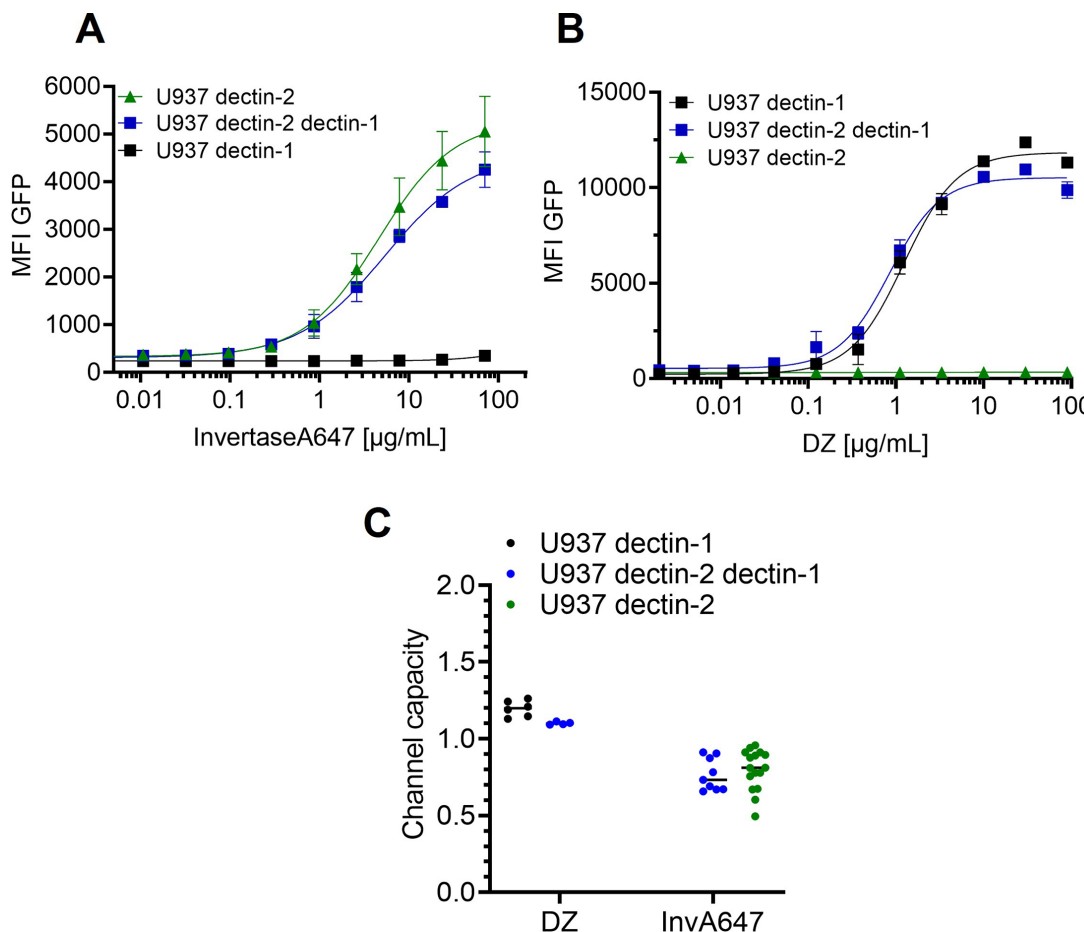

**Appendix 1—figure 3.** Signal integration of dectin-1 and dectin-2 in the presence of invertase or depleted zymosan. Monoclonal reporter cells either expressing dectin-2, dectin-1, or both dectin-2 and dectin-1 (n≥3) were stimulated for 16 hr with various concentrations of (**A**) InvertaseA647, or (**B**) depleted zymosan respectively. (**C**) Channel capacities of the data in (**A**) and (**B**).

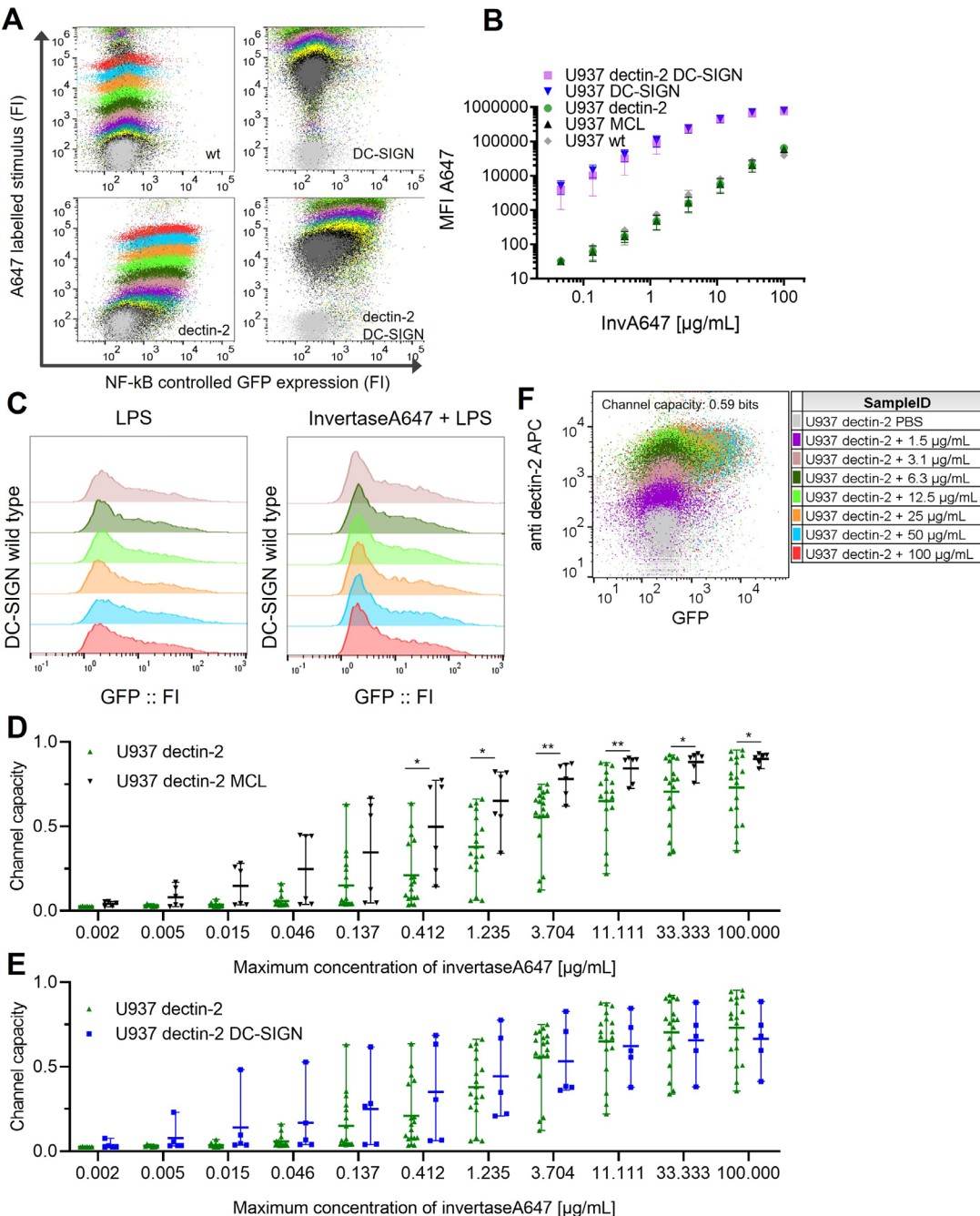

**Appendix 1—figure 4.** Signal integration between dectin-2 and either MCL or DC-SIGN in the presence of an invertase stimulus. U937 reporter cells expressing lectins as indicated, representative 2D plots (**A**) and labeled input binding (**B**) seen in the main *Figure 3B*.(**C**) U937 reporter cells either wild type or DC-specific ICAM-3–grabbing nonintegrin (DC-SIGN) expressing were stimulated with 5 µg/mL LPS-EB (invivogen) and 50 µg/mL InvertaseA647 for 18 hr. (**D**) 2D dose response and channel capacity of dectin-2 U937 reporter cells stimulated with anti dectin-2 for 16 hr. Channel capacities calculated from different maximum invertase concentrations of dectin-2 expressing cells compared with either DC-SIGN (**E**) or MCL (**F**) co-expression (*p<0.05 and **p<0.01, Wilcoxon rank-sum test).

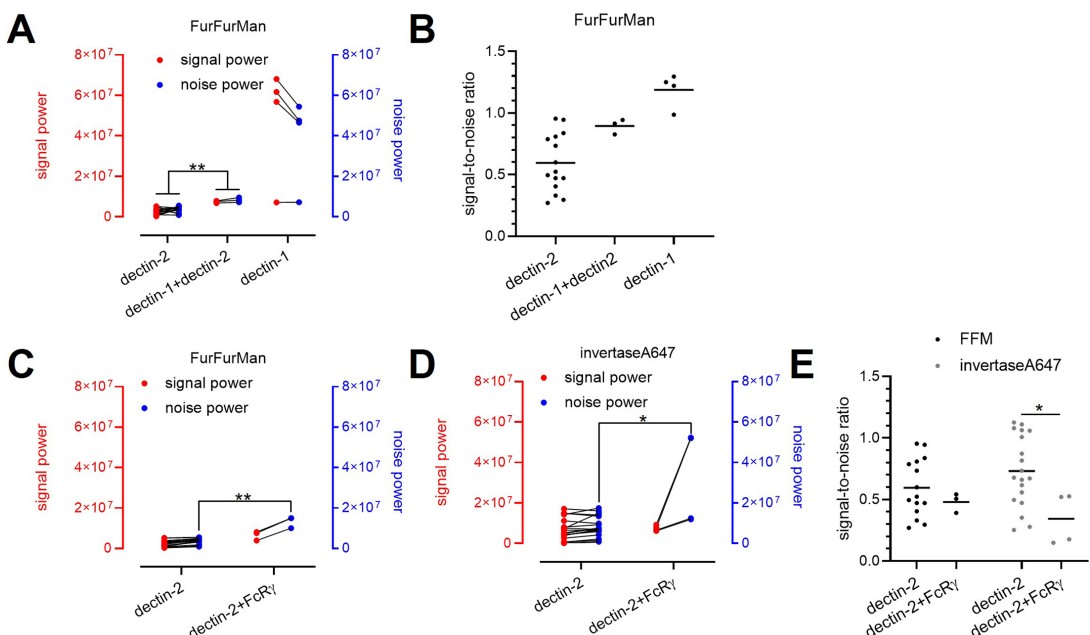

**Appendix 1—figure 5.** Decomposition of the signaling channels in various receptors and ligand conditions. (**A**) Decomposed signal (red) and noise power (blue) of dectin-2, dectin-1, and dectin-1/dectin-2 channel in the presence of FurFurMan stimulant. (**B**) Signal-to-noise ratio of (**A**). (**C** and **D**) Decomposed signal and noise power of dectin-2 and dectin-2/overexpressed FcRγ channel in the presence of FurFurMan (**C**) and invertaseA647 (**D**). (**E**) Signal-to-noise ratio (**C**) and (**D**) (*p<0.05 and **p<0.01 Wilcoxon rank-sum test).

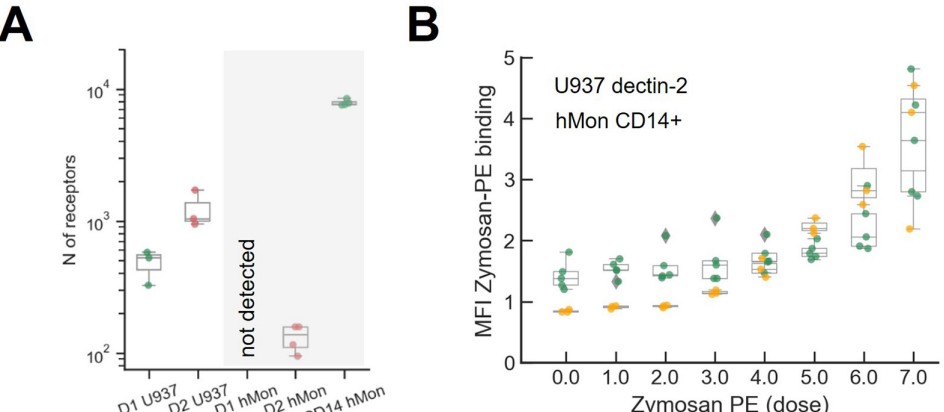

**Appendix 1—figure 6.** Quantification of the number of receptors and zymosan binding in primary cultured human monocytes. (**A**) Comparison of dectin-1 and dectin-2 expression level between our model cell (U937) and primary cultured human monocyte. The number of dectin-2 expression on U937 cells and primary human monocytes are 477±133 and 144±24, respectively. (**B**) Labeled zymosan binding on dectin-2 expressing U937 and human monocyte.

## Appendix 2

## Estimation of channel capacity between input doses and reporter GFP

### 1.1 Data structure

Cells can sense the environment and respond to it. In this work, we quantify the cellular capability to sense carbohydrate information through information theory. The carbohydrate information is given as carbohydrate ligand concentrations in the cell media, and the output is the GFP expression level triggered by NF-κB translocation of individual cells (*Appendix 2—figure 1*). The inputs (i.e. concentration of ligand) are discrete values covering almost all variability of output distribution. We used 9 or 10 levels of carbohydrate ligand concentrations including the absence of ligand and measured around 100,000 cells for whole doses using flow cytometry. The measured GFP expression levels are integer values ranging from around 0 to 50,000.

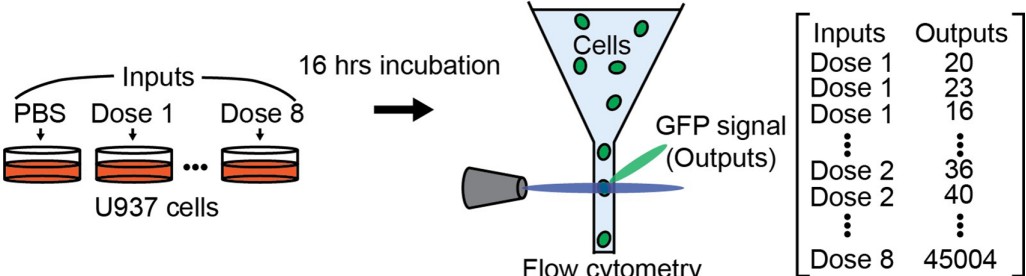

**Appendix 2—figure 1.** Illustrations showing how input and output are measured.

### 1.2 Mutual information estimation

To estimate the mutual information between carbohydrate inputs and GFP expressions from NF-κB translocation, the input and output array data described in *Appendix 2—figure 1* is projected into two-dimensional probability space divided by grids as shown in *Appendix 2—figure 2A*. The projection allows estimation of joint probability distribution of finite data points. On the other hand, finite sample size together with arbitrariness of binning interval produces over- or under-estimation of joint probability distribution, requiring additional statistical analysis to calculate unbiased channel capacity. The joint probability distribution of individual grid elements is the number of data points in the grid divided by the total number of data points. The joint probability distribution and marginalized input and output distributions is shown in *Appendix 2—figure 2 B*. The probability distribution for input and output is the marginalized joint probability distribution by output and input index, respectively, as follows:

$$P_x\left(i\right) = \sum_j^{output} P_{xy}\left(i,j\right)$$

and

$$P_y\left(j\right) = \sum_i^{input} P_{xy}\left(i,j\right),$$

where $i$ and $j$ are the input and output index, respectively, and $P_{xy}$ is the joint probability of input and output. The mutual information of the given input and output distribution is 0.53 bits, calculated from mutual information definition: $MI\left(input;output\right) = H\left(input\right) + H\left(output\right) - H\left(input, output\right)$. Note that $H\left(input\right)$, $H\left(output\right)$, and $H\left(input, output\right)$ are input entropy, input entropy, and joint entropy, respectively, defined as follows:

$$H_x = \sum_i^{input} P_x\left(i\right) \log_2 P_x\left(i\right),$$

$$H_y = - \sum_j^{output} P_y\left(j\right) \log_2 P_y\left(j\right),$$

and

$$\mathrm{H}_{xy} = - \sum_{i,j}^{\substack{input \\ and\ output}} P_{xy}\left(i,j\right) \log_2 P_{xy}\left(i,j\right).$$

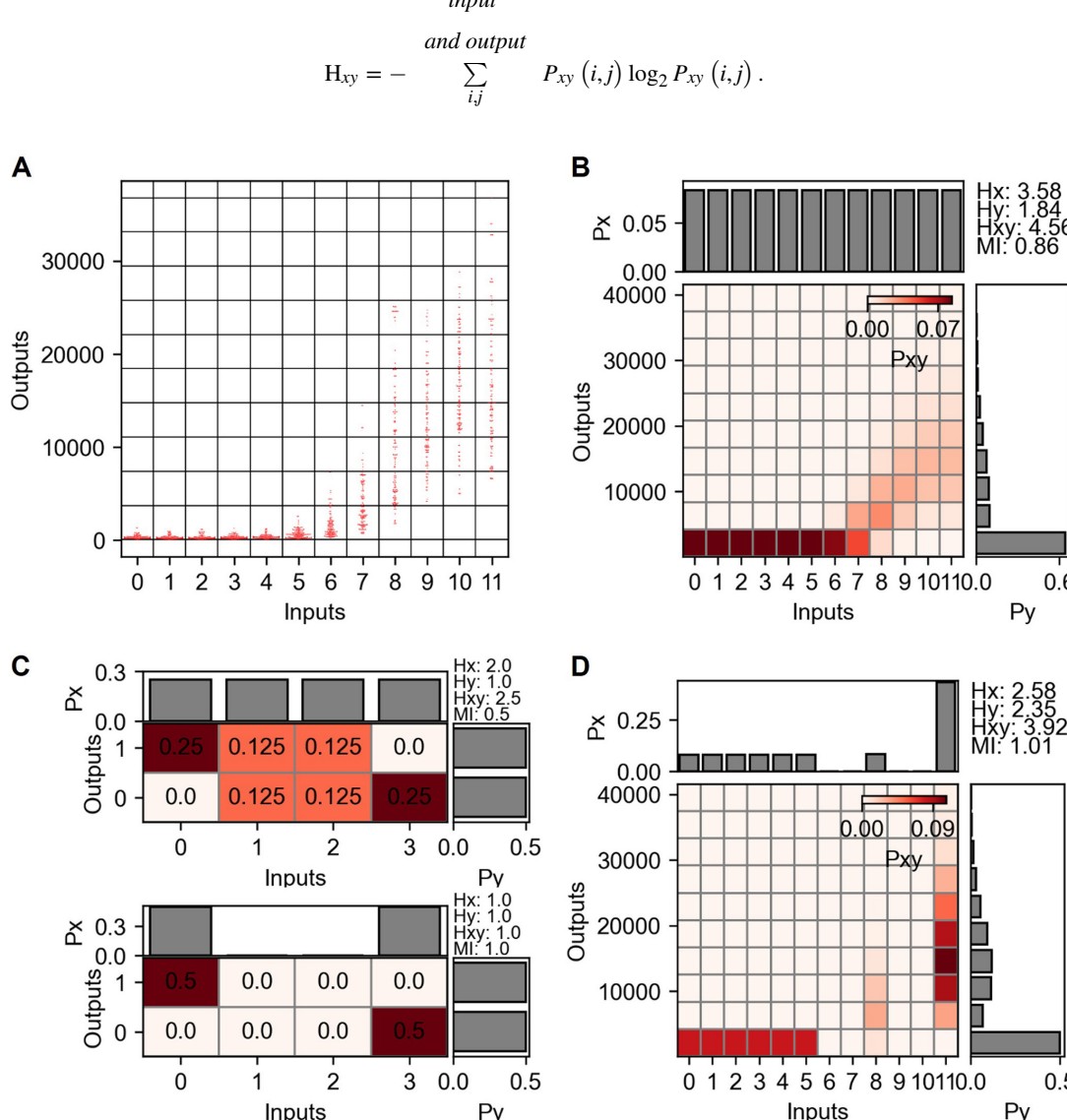

**Appendix 2—figure 2.** Mutual information and channel capacity calculation from TNF-α stimulation and nuclear factor kappa-B (NF-κB) reporter. (**A**) Grid projection of input and output distribution from dectin-2 communication channel. Inputs are the concentrations and the outputs are the GFP expression level. (**B**) Probability space describing joint and marginal probability distribution of the input and output distribution shown in (**A**). (**C**) Simple joint probability distribution consists of four inputs and two outputs signifying the channel capacity variation depending on the presence of noisy inputs. (**D**) Joint probability distribution maximizing the mutual information.

## 1.3 Channel capacity estimation

Suppose there is a joint probability distribution having four inputs and two outputs as described in *Appendix 2—figure 2C* above. Since the marginal probability distributions are equally distributed for input and output, the input entropy and output entropy yield 2 and 1 bits, respectively. The mutual information is therefore 0.5 bits by subtracting joint entropy from the sum of input and output entropy.

Consider that how much of information can be reliably transmitted from this channel? The input 0 and 3 certainly give output 0 and 1, respectively. On the other hand, the input 1 and 2 give uncertain outputs, distributed equally on all output range. Therefore, by completely suppressing the input channels 1 and 2, one can achieve the maximum information that this channel reliable transmits

(i.e. channel capacity). The array multiplication (i.e. element-wise product) between weighting values $w(i)$, [2, 0 , 0, 2], and input distribution $P_x(i)$, given in **Appendix 2—figure 2C** above, [1/4,1/4,1/4,1/4], produces the following weighted input probability distribution that maximize the mutual information:

$$P_x'(i) = w(i) \circ P_x(i) = [2,\ 0,\ 0,\ 2] \circ [P_x(0), P_x(1), P_x(2), P_x(3)] = \left[\tfrac{1}{2}, 0,\ 0,\ \tfrac{1}{2}\right].$$

And the changed input probability distribution, $P_x'(i)$, must satisfy the law of total probability

$$\sum_i^{input} P_x'(i) = \sum_i^{input} w(i) P_x(i) = 1$$

under the constraint condition

$$0 \le P_x'(i) \le 1.$$

The modified joint probability distribution yields 1 bit of channel capacity as shown in the below figure of **Appendix 2—figure 2C**.

Mathematically, the adjusted joint probability distribution by weighting values can be expressed as follows:

$$P_{xy}'(i,j) = w(i) P_{xy}(i,j)$$

And therefore the adjusted input and output marginal probability distributions are

$$P_x'(i) = \sum_j^{outputs} w(i) P_{xy}(i,j) = w(i) \sum_j^{outputs} P_{xy}(i,j) = w(i) P_x(i)$$

and

$$P_y'(j) = \sum_i^{inputs} w(i) P_{xy}(i,j).$$

Altogether, the mutual information given the weighting values is defined as

$$MI\left(input; output | w(i)\right) =$$

$$-\sum_i^{input} w(i) P_x(i) \log\ w_x P_x(i)\ -\ \sum_j^{output}\sum_i^{input} w(i) P_{xy}(i,j) \log\ \sum_i^{input} w(i) P_{xy}(i,j)\ +$$

$$\sum_j^{output}\sum_i^{input} w(i) P(i,j) \log w(i) P(i,j).$$

Finding input weighting values, $w(i)$, that maximize the mutual information subject to $\sum_i^{input} w(i) P_x(i) = 1$ and $0 \le w(i) P_x(i)$ is a non-linear optimization problem. Since the direction of the gradient of mutual information is the same as that of those two constraints at the minimum, using Lagrange multiplier method, one can restated the functions as Lagrangian $\mathfrak{L}(w_i, \lambda, \sigma) = MI\left(input; output | w(i)\right) - \lambda \left[\sum_i^{input} w(i) P_x(i) - 1\right] - \sigma \left[w(i) P_x(i)\right]$, and find out weightings using numeric approach (Kraft D. 1988. A Software Package for Sequential Quadratic Programming. Wiss. Berichtswesen d. DFVLR). We used sequential least squares programming provided by SciPy Python library (scipy.optimize.minimize, SciPy 1.7.3) to find out the optimizing input weighting values.

The input weighting values that maximize the mutual information given in **Appendix 2—figure 2B** is shown in **Appendix 2—figure 2D**. $w(i)$ is $\left[\tfrac{5}{3}, \tfrac{5}{3}, \tfrac{5}{3}, 0, 0, 0, 5, 0\right]$; hence, $P_x'(i)$ becomes $\left[\tfrac{1}{6}, \tfrac{1}{6}, \tfrac{1}{6}, 0, 0, 0, \tfrac{1}{2}, 0\right]$. Note that other weighing values, for example, $\left[5, 0, 0, 0, 0, 0, 5, 0\right]$ and $\left[0, \tfrac{10}{3}, \tfrac{5}{3}, 0, 0, 0, 5, 0\right]$, can be the solution due to the same output distribution for inputs 0, 1, and 2.

The optimized input distribution yields around 0.71 bit of mutual information between input and output distribution.

## 1.4 Channel capacity calculation from modalized weighting values

Since we use optimization algorithms, we do not predefine the weighting values to find out the maximum mutual information. On the other hand, estimating mutual information under various Gaussian-shaped input weighting values can give intuition of physiologically relevant input distribution and cellular response (*Cheong et al., 2011*).

*Appendix 2—figure 3* shows the estimation of channel capacity between TNF-α doses and NF-κB reporter under various unimodal and bimodal Gaussian input distributions. Note that several superposition of two different Gaussian distribution that forms a unimodal distribution, having single maximum peak, is excluded (*Appendix 2—figure 3 B*). *Appendix 2—figure 3 C and D*, show the calculated mutual information values from the unimodal and bimodal input distributions, respectively. Since the input range from 0 to 5 yields the same output response, the variation of input distribution within those range does not affect the mutual information. Therefore, discrete increases of mutual information are pronounced if the mutual information values are sorted in ascending order, particularly, in the case of bimodal input distribution (*Appendix 2—figure 3D*). *Appendix 2—figure 3 E and F* show the probability space given from the mutual information maximizing unimodal and bimodal input distribution, respectively. Maximum mutual information from bimodal input distribution yields around 10% higher value than that of the unimodal distribution and less than 1% of lower value compared to the optimized input distribution described in the previous section.

## 1.5 Influence of the number of output binning on channel capacity

Projection of input and output distribution onto probability space is described in *Appendix 2—figure 2 A and B*. Since the input and output data points are finite, relatively large number of output binning will produce discontinuous joint probability distribution throughout different output indexes. On the other hand, insufficient number of output binning cannot capture the original probability distribution from the input and output but average out the local variation of joint probability distribution throughout the output indexes. Therefore, the number of output binning significantly influence the mutual information and channel capacity of input and output distribution. Note that the number of input binning is the same as the number of input doses.

*Appendix 2—figure 4* describes the changing mutual information and channel capacity values in different output binning numbers. Since the input binning is given as the input doses, there is no variation in input entropy in the mutual information calculation. On the other hand, the increase of binning increases the output entropy and joint entropy. As increasing the output binning number, the increased output entropy than that of the joint entropy is bigger (*Appendix 2—figure 4 D*). Therefore, mutual information and channel capacity increase as increasing output binning.

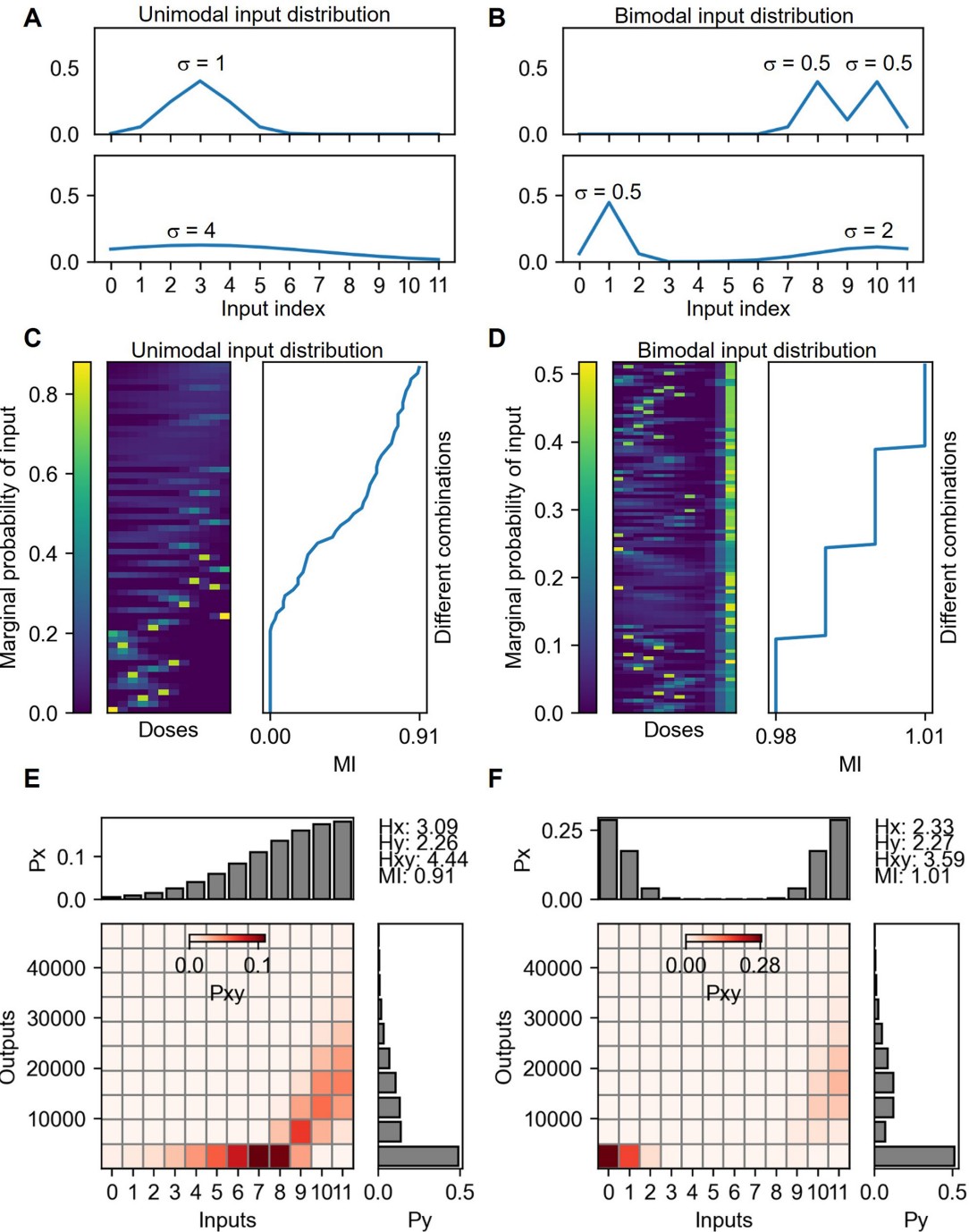

**Appendix 2—figure 3.** Mutual information calculation under unimodal and bimodal input distributions.
(**A**) Examples of unimodal input distributions. The parameter σ is the SD of the Gaussian function selected from 0.5, 1, 2, 4 and 8. There are 60 cases of input distributions. (**B**) Examples of bimodal input distribution containing the same σ parameters of the unimodal distributions. The number of bimodal combinations of the distribution is 1496. Vertically sorted various unimodal (**C**) and bimodal (**D**) input marginal probability distribution by the mutual information yields of the distribution. The probability space for the maximum mutual information given from unimodal (**E**) and bimodal (**F**) input distributions.

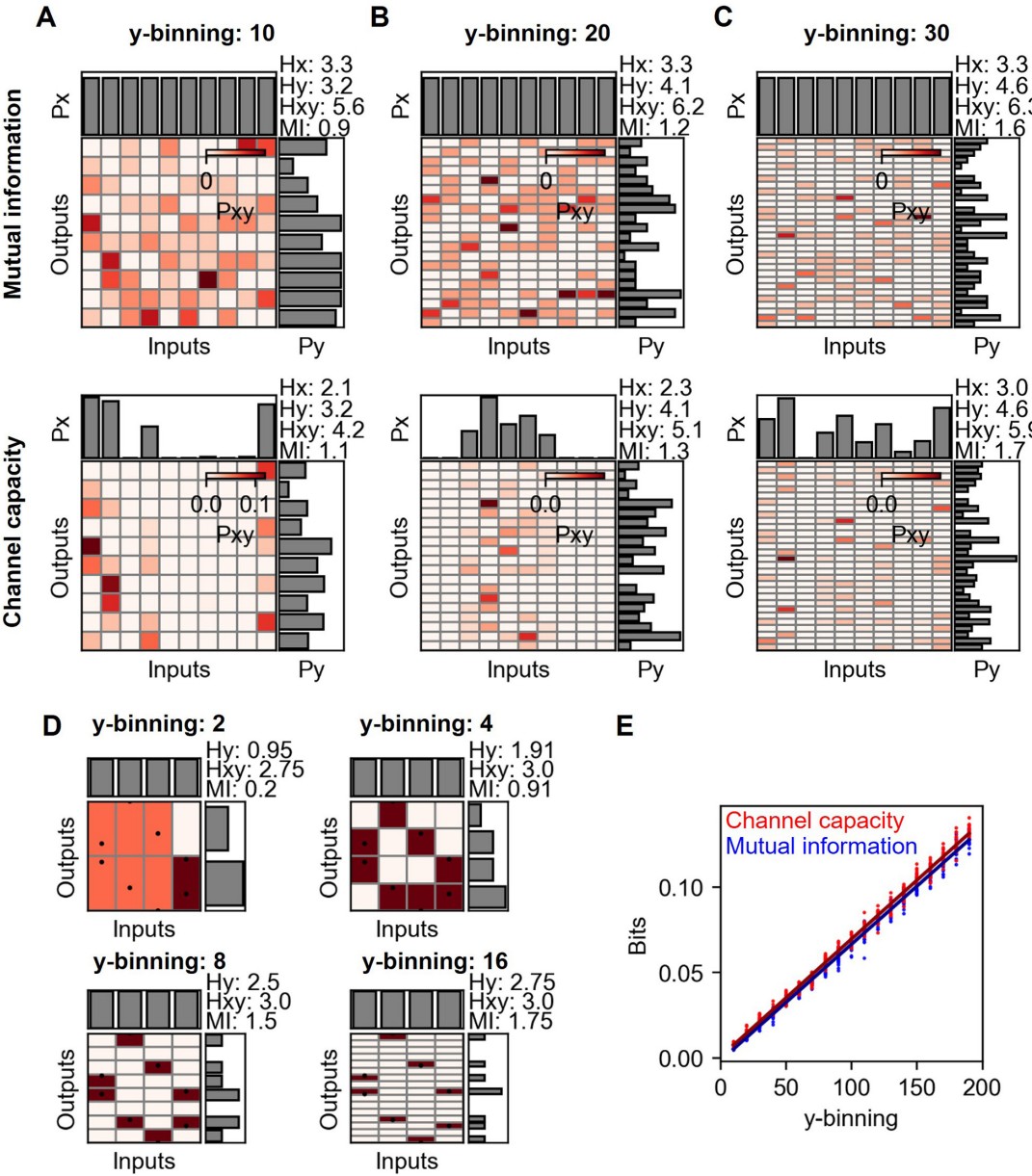

**Appendix 2—figure 4.** Mutual information and channel capacity variation depending on the number of output binning. (**A–C**) Probability space of different output binning number of random distribution (upper) and the optimized probability distribution maximizing the mutual information of the given distribution (lower). The number of random data points in each input is 10, and thereby, the total is 100. (**D**) Probability space of the same input and output distribution in different output binning. (**E**) Output binning number dependence of mutual information and channel capacity values given total 10,000 random data points distributed equally on 10 input indexes. The lines represent the linear regression.

## 1.6 Influence of the number of samples on channel capacity

As described in the previous section, it is important to consider the ratio between the number of output binning and the number of samples to estimate the channel capacity. If the number of samples is relatively smaller than the number of binning, the joint probability space become sparse and generate one to one input and output relationship which in turn increases the calculated channel capacity.

*Appendix 2—figure 5* describes changing mutual information and channel capacity with respect to the total number of samples. Since the samples are random distribution, the ground truth mutual

information and channel capacity are 0. On the other hand, the distribution yields the more mutual information and channel capacity as decreasing the total number of samples (*Appendix 2—figure 5A-C*). The mutual information and channel capacity values deviating from 0 are a bias since the ground truth mutual information and channel capacity are 0.

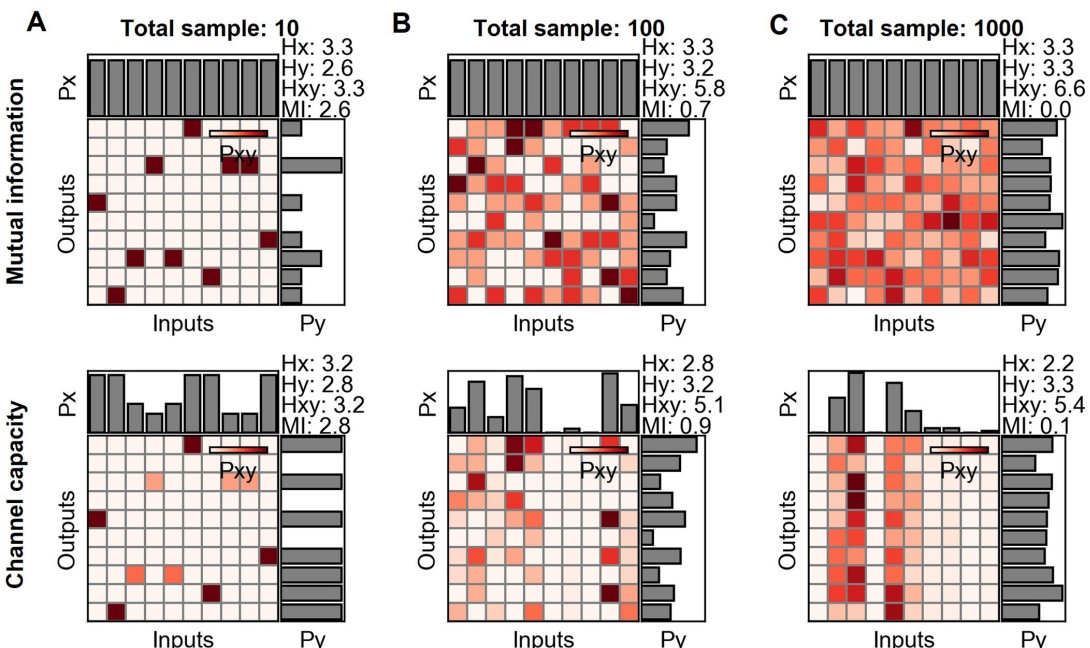

**Appendix 2—figure 5.** Mutual information and channel capacity variation depending on the sample size. (**A–C**) Probability space of different sample sized random distribution (upper) and the optimized probability distribution maximizing the mutual information of the given distribution (lower).

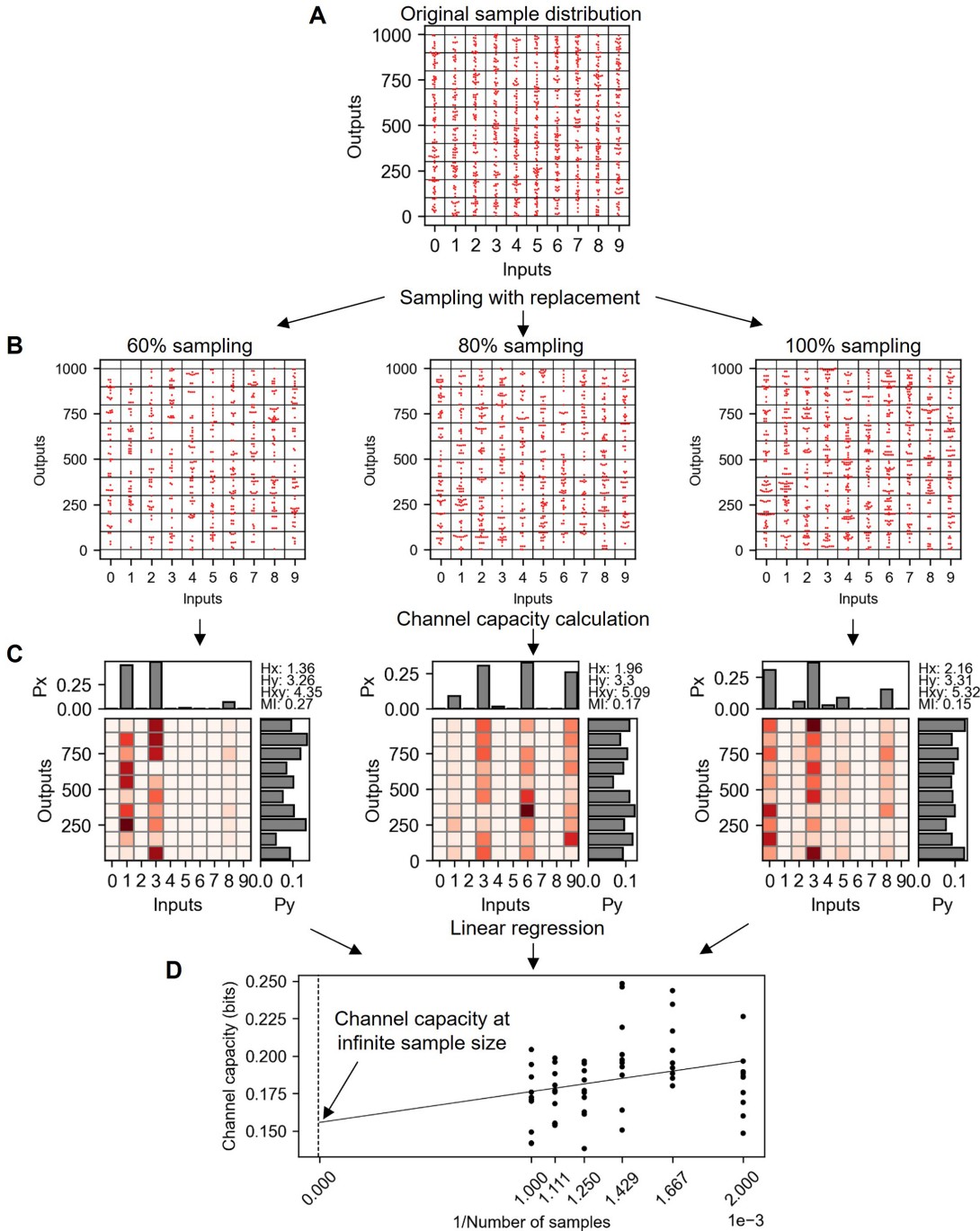

**Appendix 2—figure 6.** Explanation of bootstrapping procedure to estimate the channel capacity at infinite sample size. (**A**) Original random distribution having 100 data points in each input and therefore 1000 data points for total. (**B**) Subsampled data from (**A**) with replacement in various subsampling percentage. (**C**) Probability spaces of (**B**) maximizing the mutual information (i.e. channel capacity). (**D**) Channel capacity values calculated in various subsampling percentages. Note that the x-axis is the inverse of subsampled sample number. Therefore, the range from 0.001 to 0.002 represents the 100–50% subsampling.

## 1.7 Bootstrapping method to estimate channel capacity at infinite sample size

As shown in the previous section, the size of sample determines the degree of bias in the calculated channel capacity. Furthermore, the size of sample is always finite, therefore the calculated channel

capacity is biased. On the other hand, using linear regression of subsampled datasets, the channel capacity value at infinite sample size can be estimated as followed (*Appendix 2—figure 6*). The sample distribution given in *Appendix 2—figure 6 A* is subsampled into various subsampling percentages shown in *Appendix 2—figure 6 B*. The subsampling uses random sampling with replacement in every drawing, therefore the original sample distribution shown in *Appendix 2—figure 6* and 100% subsampled distribution from the original data show difference in distribution points (*Appendix 2—figure 6 B*). These subsampled distributions tend to yield higher channel capacity in smaller subsample size (*Appendix 2—figure 6 C*). By plotting channel capacity value with respect to the inverse of sample sizes, the channel capacity value at infinite sample size can be estimated (*Appendix 2—figure 6 D*). This bootstrapping method alleviates the degree of bias in the calculated channel capacity, but still the estimated channel capacity of random distribution at infinite sample size is around 0.15 bits, 0.15 bits higher than the ground truth channel capacity (i.e. 0 bit).

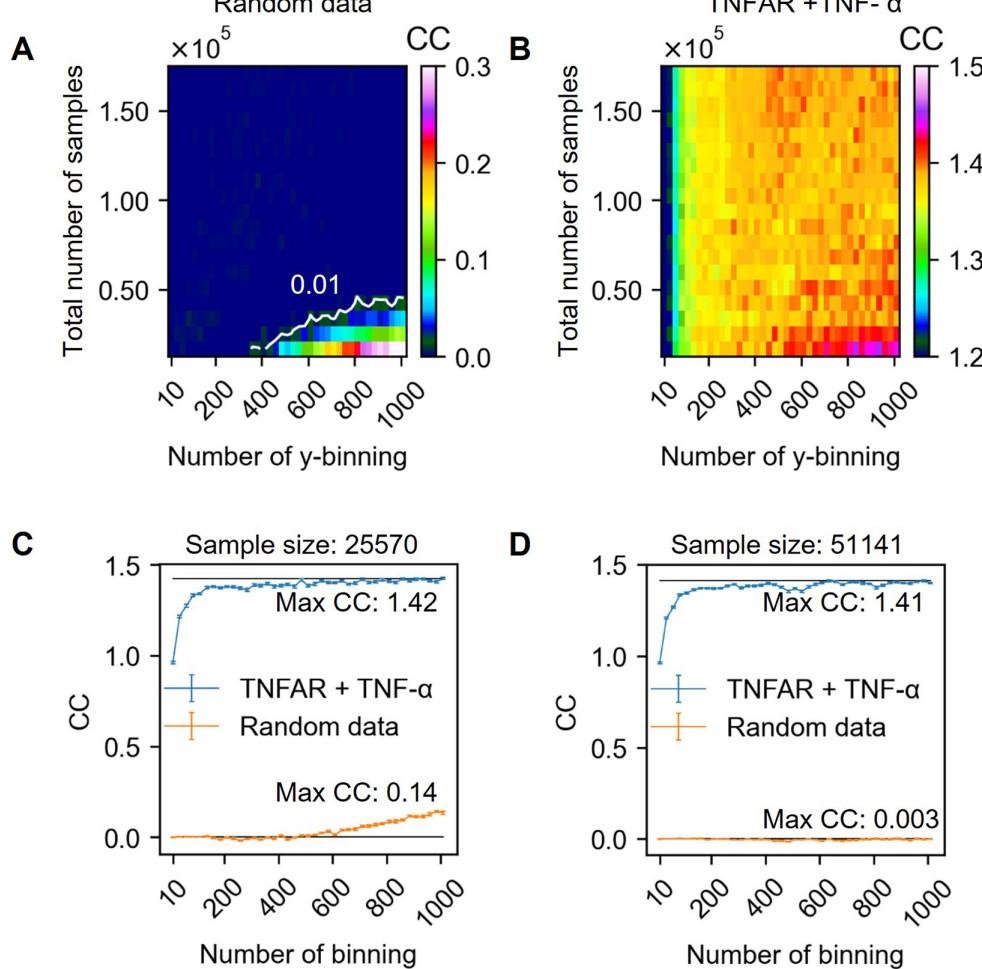

**Appendix 2—figure 7.** Channel capacity estimation in various output binning numbers and total sample numbers. (**A**) Extrapolated channel capacity values of random dataset at infinitely subsampled distribution under various total number of samples and output binning. The white line in the heatmap represents the channel capacity value at 0.01 bits. (**B**) Extrapolated channel capacity values at infinite subsample size of the input (TNF-α doses) and output (GFP reporter) of the dataset. (**C–D**) Line graphs showing the channel capacities of random and experimental dataset.

## 1.8 Channel capacity bias map depending on the output binning and total number of samples

As shown in the series of previous section, choosing an appropriate output binning and the total number of samples are essential to calculate the unbiased channel capacity. Since the two factors dependently influence the bias of channel capacity, it is required to estimate the channel capacity in

various total sample numbers and binning numbers. *Appendix 2—figure 7* above and B describe the bias map depending on the total number of samples and the number of binning calculated from random (A) and experimental dataset (B). Note that all channel capacities are estimated using bootstrapping method to interpolate the channel capacity value at infinite sample size.

In the case of random dataset, most of the regions spanning in range 0–160,000 total sample size and 0–1000 output binning number exhibit less than 0.01 bit of channel capacity. The white line in *Appendix 2—figure 7 A* indicate the contour line having 0.01 bit of estimated channel capacity. Therefore, the output binning number and total number of samples having the values above the white line exhibit less bias than 0.01 bits of channel capacity. On the other hand, the channel capacity values below the white line exhibit the value higher than 0.01 bits of channel capacity. In this work, the allowed bias is either 0.01 or 0.05 bits depending on the input and output layer (see section 123).

*Appendix 2—figure 7 B* is the bias map calculated from between TNF-α doses and reporter GFP of U937 cells. In this example, the total number of samples is 170,472 and subsampled without replacement in different y-axis of the heatmap (*Appendix 2—figure 7*). Overall, as shown in the widely spreading orange and red color, the calculated channel capacity fluctuates near 1.4 bits. The exceptions are either the case where the number of output binning is less than 200 or the coordination of total number of samples and binning numbers are below the white line shown in *Appendix 2—figure 7 A*. In this work, the minimum sample number in whole dataset is 63,816 which is the above of the line in *Appendix 2—figure 7 A*. Therefore, expected maximum bias in channel capacity is less than 0.01 bits even in the case of 1000 output binning.

We determine the channel capacity value as the highest channel capacity values calculated from output binning numbers ranging from 10 to 1000. *Appendix 2—figure 7C, D* shows channel capacity values depending on the output binning number for 25,570 and 51,141 total sample sized random distribution and Doses-GFP response data. In the case of 25,570 total sample size, around 600 output binning number, bias start to increase. On the other hand, in the case of 51,141 total sample size, there is no noticeable increase of bias in the given output binning range. The highest channel capacity values for 51,141 sample size is 1.41 bits at 985 output binning number. In the original sample size (i.e. 170,472), the maximum channel capacity is 1.41 bits at 510 output binning number. Therefore, in this example dataset, the estimated channel capacity is 1.41 bits.

In this work, if the input is discrete dose information, we estimate channel capacity using bootstrapping method described in *Appendix 2—figure 6* with multiple output binning numbers ranging from 10 to 1000. The maximum value of channel capacity in the binning range is the final channel capacity value of the calculation. *Appendix 2—figure 8* shows the results of channel capacity calculation from experimental datasets. The y-intercepts of individual lines are the unbiased channel capacity. And the maximum value of those y-intercepts in the individual dataset is selected as the final channel capacity value.

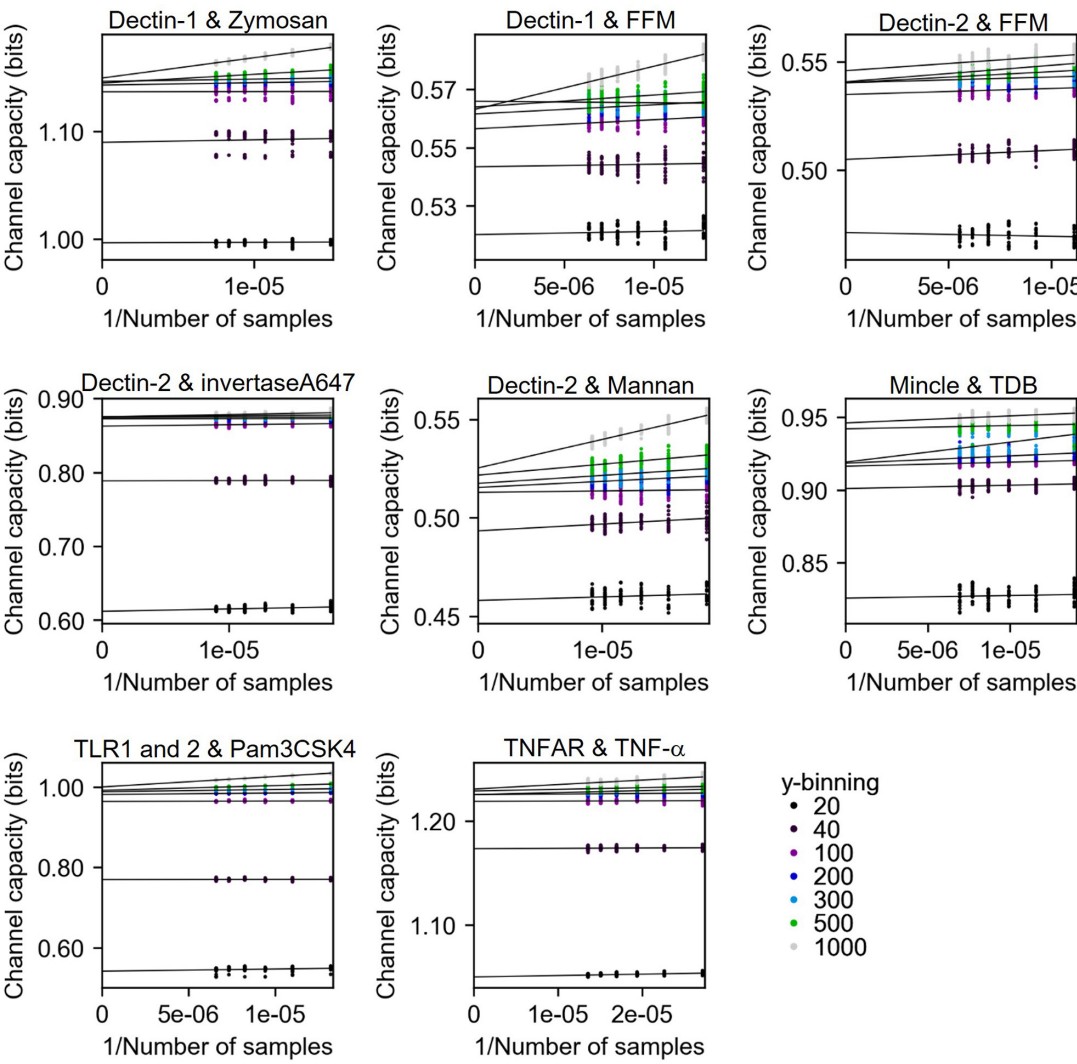

**Appendix 2—figure 8.** Channel capacity estimation of experimental data using bootstrapping in various y-binning number. The y-intercept values of the regression line are the estimated channel capacity in the given y-binning number. The number of subsampled data points in each inverse sample side is 30.

## Appendix 3

### Decomposition of signaling channel

#### 2.1 Definition of signal and noise power

The signal power of a signaling channel is the variance of the average output distribution of individual input responses. Therefore, signal power can be written as $\mathrm{E}\left[\bar{R}_i - \mathrm{E}\left(\bar{R}_i\right)\right]^2$ hence $\mathrm{E}\left(\bar{R}_i^2\right) - \mathrm{E}\left(\bar{R}_i\right)^2$, where $\mathrm{E}$ and $\bar{R}_i$ are the expectation operator and average output distribution at $i$th input dose, respectively. Since $\bar{R}_i = \sum_j R_j P\left(R_j | S_i\right)$, where $R_j$ and $P\left(R_j | S_i\right)$ are marginal output value at $j$th index and conditional probability of output at $j$th index given the $i$th input signal $S_i$, substituting $\bar{R}_i$ into $\mathrm{E}\left(\bar{R}_i^2\right) - \mathrm{E}\left(\bar{R}_i\right)^2$ provides the signal power as $\sigma_r^2 = \sum_i P\left(S_i\right)\left[\sum_j R_j P\left(R_j | S_i\right)\right]^2 - \left[\sum_{i,j} R_j P\left(R_j | S_i\right)\right]^2$, where $P\left(S_i\right)$ is the input probability at $i$th index.

In the case of noise power, it is defined as the average of the variance of the output distribution of individual input responses. Therefore, noise power can be written as $E\left[\bar{R}_i^2 - \bar{R}_i^2\right]$ and can be further expanded as $\sigma_n^2 = \sum_i P\left(S_i\right)\left[\sum_j R_j^2 P\left(R_j | S_i\right) - \left[\sum_j R_j P\left(R_j | S_i\right)\right]^2\right]$.

*Appendix 3—figure 1A, B* describes how different input and output distributions contribute to signal and noise power. Increasing the variance of output response of individual input does not influence the signal power but only increase the noise power. Likewise, increasing the mean output while keeping the variance of individual output response for each input provides increased signal power without affecting the noise power (*Appendix 3—figure 1 B*).

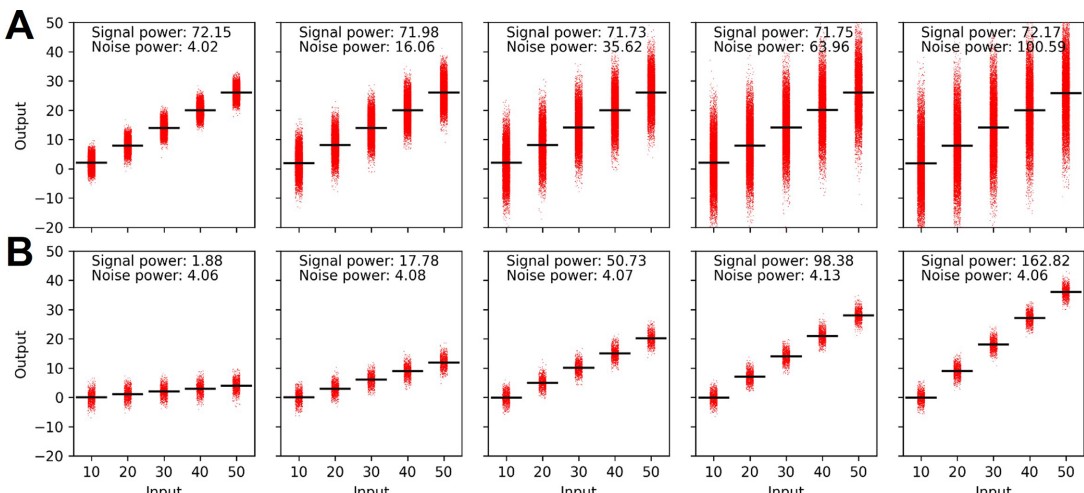

**Appendix 3—figure 1.** Signal and noise power calculated under the constraint of mean (**A**) and variance (**B**) of the output distribution.

