## [Editor Report]

This manuscript lays out the framework for addressing an important challenge in our understanding of cellular signal transduction: how complex extracellular inputs can be detected and processed using multiple receptors. This problem is addressed in the context of glycan receptors lectins, mediating very common but still not completely understood cell-cell interactions. Using information capacity analysis, the study addresses the importance of glycan input measurement by multiple receptors on the immune cells, showing how the signal detection can benefit from receptor crosstalk.

---

## [Decision Letter]

**Decision letter after peer review:**

Thank you for submitting your article "Information transfer in mammalian glycan-based communication" for consideration by *eLife*. Your article has been reviewed by 2 peer reviewers, and the evaluation has been overseen by a Reviewing Editor and Aleksandra Walczak as the Senior Editor. The reviewers have opted to remain anonymous.

Essential revisions:

1) Please, address the comments from Reviewer 1 regarding the physiological relevance of the cell model used in this study. In particular, it is imperative to relate the results to physiological levels of receptor expression and address the relevance of the signal processing in the cell type used in the experiments to cellular information transfer in vivo.

2) Both reviewers raise the key issue of how the information transfer was assessed vs previously published studies. Please, expand on the description of the information analysis, put it more explicitly into the context of the prior studies, consider supplying further details and justification for the specific capacity calculation algorithm used in the study.

3) It is important to expand on the discussion of the physiological relevance of having this complex receptor system, particularly in the context of different information transfer properties.

*Reviewer #1 (Recommendations for the authors):*

Overall, this is a fairly interesting piece of work, extending the framework of information theory to a new class of signaling networks that has not, at least to my knowledge, been considered before in this context. It is also quite interesting that there are significant differences in the information transmitted by dectin-2 and other, very closely related receptors; this suggests that even signaling systems with very similar downstream pathways can have very different noise properties. Understanding exactly why this happens mechanistically, and why these networks may have evolved this way, are very interesting future directions suggested by this work.

All of that being said, I have a number of important suggestions for the authors that I feel would need to be addressed before the manuscript could be considered for *eLife*. These are:

1) I think the physiological relevance of the results are difficult to evaluate as the manuscript is currently written. The fundamental issue is that the U937 cells considered here are not evolved to sense the glycans considered as "input" signals by the authors. Since this is an artificial cell line, the cells in question have not even evolved to sense anything; they were developed from cancer cells and have likely adapted to growing in the lab. So, while these are derived from a (potentially?) relevant immune cell population, they are clearly not representative of the actual human cells that have evolved to sense and respond to potential fungal infections. As far as I can tell, these cells don't even express the relevant receptors, since the authors have to introduce those receptors exogenously.

The authors may argue that the cells in question are at least immunological in origin, and that cell line work of this type is common in the field due to the difficulty of engineering the appropriate reporters into, say, a mouse in order to do work in primary cells. That may be true, but it still does not alleviate a very critical set of concerns regarding these results. In particular, it is currently difficult to interpret the findings the authors have about dectin-2 having a lower 'channel capacity' than other receptors like dectin-1. As the authors (basically) state, there are two hypotheses regarding this particular result. It does not seem that the difference in channel capacity between dectin-2 and dectin-1 is due to differences in average expression levels or due to differences in the distribution of protein levels themselves. Indeed, compared to TNF-α, there actually seems to be less noise in dectin-2 binding vs. TNF-α binding (more on that question below, see Figure 4A). So, the differences in channel capacity here are likely either due to fact that dectin-2 is somehow just more "noisy" as a receptor in and of itself, or the fact that there are differences in the downstream signaling network in between the receptor and NFκB.

This second hypothesis seems far more likely, given that it is highly unclear that intrinsic differences in, say, the conformational changes that drive signaling processes downstream signaling could be that significantly different between dectin-1 and dectin-2. In other words, it seems extremely unlikely that an intrinsic, biophysical difference between the receptors could be sufficient to explain the difference in channel capacity. As such, it is likely that the real difference here is due to some difference in downstream interactions. The authors seem to imply that any such differences are currently unknown, since these two receptors seem to interact with the same downstream pathways. Regardless, it seems much more likely to me that differences in the signaling networks induced by these two receptors (or dectin-2 and any of the receptors studied here) are driving differences in the observed information transfer.

It has been shown that the channel capacity depends in a critical way on the distribution of the number of downstream signaling molecules present (see the Suderman et al. (2018) Interface Focus 8 (6), 20180039). In other words, if you have a network that signals through a molecule with low abundance, that will tend to increase noise levels and decrease channel capacity. Or, if there is a molecule in the signaling pathway with a particularly broad distribution across the cell population, that can also lead to low channel capacities.

The problem in this case is that the cells in question have not evolved to use dectin-2 to sense anything in the environment. So, a particular protein downstream of dectin-2 may be expressed at a low level, or expressed with a broad distribution, simply because there is no need for these cells to sense their environment efficiently. As a result, it is very difficult to interpret the physiological relevance of the results presented here.

As currently written, the authors seem to make the argument that their findings have revealed a fundamental difference between dectin-2 and the other signaling molecules considered here. I am not myself convinced of that fact, simply because the physiological relevance of exogenously expressing these receptors in an immortal cancer cell line is unclear. I would suggest the authors think about what their results truly mean for our understanding of the underlying biology. Currently, I think it is impossible to infer that the authors would see the same kind of results in primary cells that actually perform the job of sensing these molecules in the body. That being said, the results are still interesting-dectin-1 and dectin-2 are very closely related, and to have such different noise properties is really rather intriguing. It suggests that receptors can have very different dose-response distributions even if they are very similar and are thought to signal through (essentially) identical downstream systems. The fact that this can happen is surprising, I think, and sets up the two alternative hypotheses discussed above. But the authors need to acknowledge this and be more reasonable regarding the extent to which these results can be extrapolated into a more meaningful biological context.

2) One of the most interesting aspects of this work is the fact that the authors consider not just the downstream signaling response, but also directly measure the statistical properties of the first step in the signaling process: namely, receptor binding. I think there is a significant opportunity to understand a critical aspect of cell signaling here in a new and interesting way, and so I have some suggestions for analyses the authors could do that would, I think, really increase the impact of the work.

I would be really interested to see what the channel capacity would be for the information flow between ligand concentration and the level of bound receptor. In other words, the authors can calculate the channel capacity between the dose of the ligand that they give the cells, and the distribution of bound receptor that they observe on the cell surface.

Looking at the data (e.g. Figure 4A), my intuitive sense is that the channel capacities will be relatively low for that calculation: maybe 2 bits or something like that. Interestingly, that is much, much lower than the theoretically possible value we would expect for receptors expressed at around 1000 copies per cell (which is around 3.2-4 bits, see the Suderman Interface Focus paper mentioned above). This may simply be because the receptors are being expressed exogenously and thus have lower absolute numbers, and broader distributions across the population, than would be observed in a more physiological context. That being said, this is to my knowledge the first actual measurement of the amount of bound receptor on the surface of individual cells. This is thus a great opportunity to calculate the information flow between ligand concentration and bound receptor levels.

I should note here that it might be intuitive for the authors to see this as an "upper bound" on the channel capacity for their system; in other words, the channel capacity between the ligand concentration and the bound receptor concentration should set an upper bound for the channel capacity for the entire system. The data processing inequality from Shannon seems, at least on the face of it, to guarantee that. It is important to note, however, that the signaling networks in question do not form a Markov chain; the amount of reporter expression can easily average over the entire signaling history in a particular cell. So, whatever the authors find here, it is not technically an upper bound on the downstream information flow. That being said, it would still be very interesting to know!

It should be relatively easy for the authors to just make this calculation based on the data they already have. I do, however, have suggestions for a few experiments that would improve the rigor of this calculation and make the results even more interesting.

Firstly, it is unclear how representative the bound protein numbers are of the actual number of binding events we would have in the cell culture when the cells are exposed to the ligand. This is because the authors have to wash the cells (at least I assume they do that) and then put them through the FACS instrument to measure the fluorescence. It would be really cool of the authors could do a time course to measure how fast the ligand "falls off" of the receptor. As long as the measurements made by the authors occur before significant unbinding has happened, the channel capacity they calculate should be very representative of what actually happened in the culture.

The other experiment that would be really interesting would be to repeat the experiment with labeled ligands but using whatever primary cells are known to express dectin-1, dectin-2, etc. In other words, the fluorescence here is coming from the ligand, so there is no need to engineer any reporters into the cell. If the authors see similar distributions of bound receptor on the surface of primary cells, and similar channel capacities between ligand concentration input and the bound receptor as output, that would be very intriguing. In any case, doing this experiment would I think begin to address some of the concerns with physiological relevance raised above. It would also be extremely interesting as a contribution to the field.

3) As mentioned in my public comments, another issue in this work is the fact that the authors are relatively reticent about how their calculations are actually performed. While the authors present the relevant equations for calculating the Mutual Information and claim to follow the approach of Cheong et al., the methods are not sufficiently detailed to understand exactly what they have done here. This is particularly important because their approach is clearly not identical to that used by Cheong, since their algorithm for maximizing the MI across input distributions is different. Cheong et al. approach this problem by trying a limited set of possible input distributions and using the one that gives the highest MI, while the authors here use a built-in optimization algorithm in python. It is not clear if this is the only point at which the authors deviate from the approach laid out by Cheong et al., or if they have modified other aspects of the calculation.

The authors need to explain how they approach the problem of correcting for finite size effects. Do they use the approach of Cheong et al., bootstrapping the data at different levels and then using a linear extrapolation of MI vs. 1/N (where N is the number of data points/cells) to extrapolate to the case of an infinite population size (1/N \to 0). Did they use the same approach to choosing the number of bins to calculate the probability distribution over outputs? Cheong et al. did this by visual inspection, if I recall correctly, looking for a "plateau" in bin numbers where the MI calculated from the data was non-zero but the MI calculated for randomized data was 0. Based on the statements made by the authors, and Figure S6, they seem to have performed both of these steps-otherwise it is completely unclear where the confidence intervals on their MI/channel capacity estimates come from, and how they chose the number of bins to use. Although it is important to note that the randomized controls seem to be missing from Figure S6. Regardless, since the authors have evidently re-implemented the code, they need to explain exactly what they have done.

I would suggest that the authors also adopt several improvements to the Cheong et al. approach that have been developed in subsequent works. In particular, Suderman et al. (2017, PNAS 114 (22), 5755-60) implemented a broader range of bootstrap samples across which to do the linear extrapolation to infinite population size, automated the choice of bin sizes, and developed a bootstrap approach to estimating confidence intervals that is much more realistic (and statistically appropriate) than simply using the confidence intervals from the linear extrapolation step. The authors can refer to the Suderman paper on how this was done; it should be relatively easy to implement these improvements, if they have not been done already.

I appreciate the author's innovation of using an optimizer in python to maximize the MI. I would expect this actually produces higher "channel capacities" than the previous approaches mentioned above. That being said, it is unclear how this optimization algorithm works. The authors may expect readers to go to the python documentation, but there are problems there. For one, the algorithm implanted in python might change, so the documentation may describe an approach that is different from the one the authors actually used to perform their calculations. How various packages are implemented in python can change from time to time, and something so generic as "optimize" may be changed as better optimization algorithms become available in the future. So the authors should describe exactly what they did. Also, readers are, in general, not going to go to python documentation to understand the methods of a paper like this, nor should they have to. The authors should describe what the algorithm is doing as currently implemented, in terms that readers can readily understand.

Finally, the authors should also perform the calculation using the exact same distributions used by Cheong et al. and Suderman et al. This will allow us to compare their results more directly to the results from those previous authors, as well as allow us to determine if the approach that they used results in similar channel capacities to the optimization algorithm used here. While I expect the optimization algorithm employed by the authors will yield larger numbers for the channel capacity (and thus better estimates, since the channel capacity is formally a supremum), it would be extremely useful if the authors checked this.

4) Another issue related to the calculation of the MI itself is the fact that the authors use logarithmically spaced bins, rather than linear bins, to perform the calculation. The authors describe this approach as if the matter of how to construct the bins is simply a matter of personal preference or convenience for the calculation, and choose logarithmic bins because they give somewhat larger values for the channel capacity. For one, this is not surprising; as is often the case, the FACS data obtained by the authors has a log-normal-like character; in other words, the authors see normal-ish distributions on a log scale (e.g. Figure 2A). As such, it is natural that using logarithmic bins will result in higher channel capacities, because it will tend to equalize (as much as possible) the effective number of observations within the bins.

While this may seem reasonable at first, it is actually, in my view, hard to justify. The reason is that the choice of how to generate the bins is not arbitrary, but actually corresponds to a very strong statement regarding what the authors actually consider the relevant "output" of the system to be. Using logarithmic bins is obviously equivalent to first taking the log of the data, and then using linear bins on that log-transformed data (Figure S2A should make this fact abundantly clear). The authors are thus envisioning that the channel is not a channel whose input is ligand concentration and output is GFP level, but rather a channel where the input is the ligand concentration and the output is the log of the GFP level. This is not an arbitrary decision, but rather one that has real consequences for how we construe the calculation in a physiological context. If the output is the log of the GFP level, that means that the cell is sensitive not to linear changes in protein concentration, but rather fold-changes.

For certain transcription factors, authors have argued that it is actually this kind of relative, fold change that matters (see Lee et al. (2014) Mol Cell 53 (6) 867-79). I would argue, however, that the conclusions of that work are hardly so robust as to suggest that, for every input-output transcriptional regulatory system within cells, it is the fold change of the "output" protein, rather than its linear change, that matters to the cell. Certainly, if the output is an enzyme, a transporter, or a protein that performs a whole host of other functions, then it is natural to assume that the appropriate output is the protein concentration itself, not the fold change in protein concentration.

If the authors can argue that every single gene whose transcription is controlled by NFκB has a fold-change impact on its downstream function, then I could see a strong argument being made for logarithmic bins. As it stands, however, I think the most natural way to interpret the output of the channel is on a linear scale, and I would say the authors should focus on that, rather than a logarithmic scale. If the authors want to include their logarithmic calculations in this work, I would suggest moving them to the supplement, and making clear the fact that such a calculation construes the output as having a fold-change impact on whatever is downstream of the protein level measured by the authors.

5) In this work, the authors focus on the measurement of GFP levels at a certain time. As written, it is not clear if the authors are considering the steady-state response of the cells that they are treating, a peak response, or something else. As far as I can tell, the authors stimulate the cells for 16 or 13 hours, depending on the ligand in question. I am not sure how these numbers were chosen-did the authors look at the GFP expression dynamics and choose this number based on those measurements? From an information-theoretic standpoint, it is best if the results are based on steady-state protein levels, but the author's don't seem to explain their rationale for choosing the time points that they choose. If the GFP expression levels do not reach a steady-state, it would be great if the authors had a particular reason for choosing the times they choose. I would also suggest that the authors consider looking to see if their channel capacity estimates are robust to the time they choose by looking at other time points and calculating the channel capacities to see if they get similar results. I am not sure what times to choose, but perhaps 24 hours or 36? It is hard to say without looking at the average dynamics over time, so providing that kind of data would be extremely helpful.

Also, the authors should acknowledge that a large body of literature has emerged that makes the claim that it is not the individual time points that matter, but rather the entire time series of the response that should be construed as the output. The authors can refer to the Zhang et al. paper in Cell Systems, the paper by Selimkhanov et al. (Science (2014) 346 (6215) 1370-3), papers by the Hoffman group (notably Tang et al. (2020) Nat Commun 12 (1) 1272), and a host of others. I am not suggesting that the authors adopt this worldview; there are actually serious mathematical and conceptual issues with construing the output of a communication channel to be a function like a time series. But, this idea is out there in the literature, and the authors should address this point and discuss how looking at GFP time series, rather than individual time point measurements, might yield different (and undoubtedly higher) channel capacity estimates.

5) As a final, rather important but also rather technical point, the authors in Figure 3C do statistical tests on their channel capacity estimates, comparing dectin-1 and dectin-2 cells to those that express dectin-1 and dectin-2 together. The issue here is that they use a t-test to estimate statistical significance, but it is really unclear where the dispersion (e.g. error bars or standard deviations) in the channel capacity estimates come from. I expect, since the authors day "n = 9" in the legend, that they split their data into 9 groups (maybe 9 sets of cells whose results were collected in different FACS runs or something?), performed the channel capacity estimate on each independently, then used the means and standard deviations in those estimates to do the t-test. That is completely made up by me, however; the authors really need to explain where these error estimates (and their +/- X values for channel capacity estimates in the various tables) actually come from.

In any case, the t-test of course only works if the data in question is drawn from actual Gaussian distributions. This does not mean that the distributions "look Gaussian" or "seem okay;" in the application of a t-test, you make extremely strong assumptions that only real honest-to-goodness Gaussian distributions actually satisfy. So the data really needs to actually be drawn from a Gaussian distribution for the results of the test to be interpreted appropriately.

My first suggestion here is that the authors need to use a non-parametric statistical test. That is, unless there is a true, theoretical reason to expect that the data will be drawn from a Gaussian (I know of no such reason, but perhaps the authors have one). The Wilcoxon rank-sum test was made for problems like this one, so I would suggest the authors use that.

A larger problem here, however, is that taking n = 9 (however it was done in the end) is not going to give a great estimate of the uncertainty in the channel capacity estimate. I would instead suggest that the authors merge their data into one data set (unless there are some kind of batch effects that prevent them from doing that) and then using a bootstrap/resampling approach, as in the Suderman et al. PNAS paper, to estimate confidence intervals and generate distributions for statistical tests. These bootstraps can be done hundreds or thousands of times, allowing for even more non-parametric tests (like permutation tests) to be used to estimate statistical significance.

This is honestly a fairly minor point, because nothing terribly important in the conclusions of the paper rest on the statistical tests here. But, if the authors want to argue that there are real differences between dectin-1, dectin-2 and dectin-1+dectin-2 cells, they should do these statistics in a more non-parametric way.

*Reviewer #2 (Recommendations for the authors):*

I think this paper would be greatly improved if the authors were more precise and detailed in the description of their capacity calculation. To address this, it's likely the authors were following the method from Cheong et al., but some of those details should be described here.

---

## [Author Response]

Essential revisions:1) Please, address the comments from Reviewer 1 regarding the physiological relevance of the cell model used in this study. In particular, it is imperative to relate the results to physiological levels of receptor expression and address the relevance of the signal processing in the cell type used in the experiments to cellular information transfer in vivo.

We thank the reviewer for their concern and updated the revised manuscript accordingly. We agree that for future work such a correlation between model cell lines and primary cells will benefit most of the biophysical and synthetic biology work focused on the interpretation of the signaling transmission. However, direct implementation of the NF-κB reporter into primary cells is very challenging. Hence, we decided to validate the choice of the model cell line by direct, side-by-side, comparison of dectin-1 and dectin-2 positive U937 reporter cells with the primary human monocytes and have mentioned and included these data in manuscript (line 382) and the Supporting Information (Appendix 1 Figure 6), respectively. These experiments revealed that human monocytes to the large extent overlap with the dectin-2 positive U937 cells in their expression level of dectin-2 receptor and efficiency to interact with zymosan particles.

Additionally, at this early stage of our understanding for the complexity of glycan-mediated information transmission, we see several advantages of our approach: (i) U937 cells originate from human monocyte and expresses immunological receptors such as mincle, CD200, CD200R, Siglec-1, and Siglec-3, therefore resemble physiological relevance (Byrareddy et al. PLOS ONE 10, e0140689 (2015)). Model cells must contain all necessary down streaming molecules connecting our receptors of interest to NF-κB. By this the model cell line we chose here will also be applicable for future studies on other glycan binding proteins associated with the immune system. (ii) We used monoclonal U937 cells, thus our results are independent from the heterogeneity of cells.

On the other hand, we agree that the U937 cells may differ in their expression levels of these downstream components of the signaling pathways and in turn, influence the activation of NF-κB, which may alter the channel capacity. Hence, in the revised manuscript, we do not compare the channel capacity between two different receptors that have different downstream pathways such as TNF-a and dectin-2.

2) Both reviewers raise the key issue of how the information transfer was assessed vs previously published studies. Please, expand on the description of the information analysis, put it more explicitly into the context of the prior studies, consider supplying further details and justification for the specific capacity calculation algorithm used in the study.

In the revised manuscript, we added a general introduction on channel capacity calculation method that used in this study and its comparisons to prior studies as described in Appendix 2 and 3.

3) It is important to expand on the discussion of the physiological relevance of having this complex receptor system, particularly in the context of different information transfer properties.

Thank you for raising this concern. We agree that the physiological relevance needs to be discussed further and we expanded the Discussion to explain why we choose U937 and its physiological relevance as follows:

Line 382: “Finally, it is important to take into consideration that our conclusions came from model cell lines, which were used as a surrogate for cell-type-specific lectin expression patterns of primary immune cells. Human monocytes and dectin-2 positive U937 cells have comparable receptor densities and respond similar to stimulation with zymosan particles (Appendix 1 Figure 6A and B).”

Reviewer #1 (Recommendations for the authors):Overall, this is a fairly interesting piece of work, extending the framework of information theory to a new class of signaling networks that has not, at least to my knowledge, been considered before in this context. It is also quite interesting that there are significant differences in the information transmitted by dectin-2 and other, very closely related receptors; this suggests that even signaling systems with very similar downstream pathways can have very different noise properties. Understanding exactly why this happens mechanistically, and why these networks may have evolved this way, are very interesting future directions suggested by this work.All of that being said, I have a number of important suggestions for the authors that I feel would need to be addressed before the manuscript could be considered for eLife. These are:1) I think the physiological relevance of the results are difficult to evaluate as the manuscript is currently written. The fundamental issue is that the U937 cells considered here are not evolved to sense the glycans considered as "input" signals by the authors. Since this is an artificial cell line, the cells in question have not even evolved to sense anything; they were developed from cancer cells and have likely adapted to growing in the lab. So, while these are derived from a (potentially?) relevant immune cell population, they are clearly not representative of the actual human cells that have evolved to sense and respond to potential fungal infections. As far as I can tell, these cells don't even express the relevant receptors, since the authors have to introduce those receptors exogenously.The authors may argue that the cells in question are at least immunological in origin, and that cell line work of this type is common in the field due to the difficulty of engineering the appropriate reporters into, say, a mouse in order to do work in primary cells. That may be true, but it still does not alleviate a very critical set of concerns regarding these results. In particular, it is currently difficult to interpret the findings the authors have about dectin-2 having a lower 'channel capacity' than other receptors like dectin-1. As the authors (basically) state, there are two hypotheses regarding this particular result. It does not seem that the difference in channel capacity between dectin-2 and dectin-1 is due to differences in average expression levels or due to differences in the distribution of protein levels themselves. Indeed, compared to TNF-α, there actually seems to be less noise in dectin-2 binding vs. TNF-α binding (more on that question below, see Figure 4A). So, the differences in channel capacity here are likely either due to fact that dectin-2 is somehow just more "noisy" as a receptor in and of itself, or the fact that there are differences in the downstream signaling network in between the receptor and NFκB.This second hypothesis seems far more likely, given that it is highly unclear that intrinsic differences in, say, the conformational changes that drive signaling processes downstream signaling could be that significantly different between dectin-1 and dectin-2. In other words, it seems extremely unlikely that an intrinsic, biophysical difference between the receptors could be sufficient to explain the difference in channel capacity. As such, it is likely that the real difference here is due to some difference in downstream interactions. The authors seem to imply that any such differences are currently unknown, since these two receptors seem to interact with the same downstream pathways. Regardless, it seems much more likely to me that differences in the signaling networks induced by these two receptors (or dectin-2 and any of the receptors studied here) are driving differences in the observed information transfer.It has been shown that the channel capacity depends in a critical way on the distribution of the number of downstream signaling molecules present (see the Suderman et al. (2018) Interface Focus 8 (6), 20180039). In other words, if you have a network that signals through a molecule with low abundance, that will tend to increase noise levels and decrease channel capacity. Or, if there is a molecule in the signaling pathway with a particularly broad distribution across the cell population, that can also lead to low channel capacities.The problem in this case is that the cells in question have not evolved to use dectin-2 to sense anything in the environment. So, a particular protein downstream of dectin-2 may be expressed at a low level, or expressed with a broad distribution, simply because there is no need for these cells to sense their environment efficiently. As a result, it is very difficult to interpret the physiological relevance of the results presented here.As currently written, the authors seem to make the argument that their findings have revealed a fundamental difference between dectin-2 and the other signaling molecules considered here. I am not myself convinced of that fact, simply because the physiological relevance of exogenously expressing these receptors in an immortal cancer cell line is unclear. I would suggest the authors think about what their results truly mean for our understanding of the underlying biology. Currently, I think it is impossible to infer that the authors would see the same kind of results in primary cells that actually perform the job of sensing these molecules in the body. That being said, the results are still interesting-dectin-1 and dectin-2 are very closely related, and to have such different noise properties is really rather intriguing. It suggests that receptors can have very different dose-response distributions even if they are very similar and are thought to signal through (essentially) identical downstream systems. The fact that this can happen is surprising, I think, and sets up the two alternative hypotheses discussed above. But the authors need to acknowledge this and be more reasonable regarding the extent to which these results can be extrapolated into a more meaningful biological context.

We agree on Reviewer #1’s comment regarding the difference of cell type can influence the amount of signaling molecules and thereby the channel capacity. Therefore, there is an arbitrariness in the calculated channel capacities. Indeed, we made model cell lines in THP1 and showed different NF-κB response but relatively weak compared to U937 cell. At this point we can only speculate why this is and assume that U937 cell has more downstream signaling molecule compared with the other cell lines. In the corrected manuscript, we do not compare the channel capacity between two different receptors that have different downstream pathway such as TNF-a and dectin-2.

The expression level of various downstream molecules of U937 cell might be different to the macrophages or dendritic primary cells. But U937 has all essential signaling molecules to translocate NF-κB with our receptors of interest. Therefore, we could investigate how two different carbohydrate receptors compromise the incoming carbohydrate information to the NF-κB response; This is hardly done by primary cells.

To address the concern, as was suggested, in the updated version of the manuscript we directly compared the receptors expression level of CD14+ human monocytes primary cells and our model synthetic cell lines (Appendix 1 Figure 6). Overall, it is challenging to introduce NF-κB reporter in the primary cells and one advantage of our system is that U937 reporter cell line was expanded from a single clone. Thus, our results are not affected by the cell-to-cell variability, providing more defined mechanistic conclusions. Since we could not introduce the reporter into the primary cells, we correlated the response of dectin-1 and dectin-2 positive U937 cells with the primary classical monocytes by comparing: i. binding of glycan input and ii. quantification of the receptors number. As a result, we observed that dectin-2 positive U937 cells and classical monocytes can interact comparably with the labeled glycan input. We also found that the receptor density on the transduced cell line was one order of magnitude higher in comparison to primary monocytes. We believe that such discrepancy may dependent on the primary cell type (unfortunately we could not run similar studies on multiple immune cell samples (1000€ each)), or variability in their isolation protocol. Thus, we consider the choice of our model system physiological and relevant for the behavior of classical human monocytes, at least based on the parameters we quantified in this direct side-by-side comparison.

We also would like to emphasize that our conclusions on the noisiness of dectin-2 receptor are coming from the direct comparison between dectin-2 and mincle receptors. Although dectin-1 and dectin-2 are receptors of the same lectin family, they have low structural homology and completely different mechanism of the downstream signal transmission and originate from different gen clusters (Saijo and Iwakura, 2011). One major difference for instance, is the presence of the ITAM-domain on dectin-1 and the absence of such on dectin-2. In addition, mincle cannot be phosphorylated on its own, similar to dectin-2, and relies on the downstream interaction with FcRy receptor. Thus, we also agreed with the review, that the different noise for signaling via these two receptors is an interesting observation which deserves further focus. This is ongoing research in the lab.

Currently, we assume that the noisiness of dectin-2 might be hidden in the sequence of its intracellular domains. In the follow-up work we plan to swap intracellular sequences of dectin-2 and mincle and check how the construction of such chimera receptors affects signaling efficiency. To summarize, in our work we concluded on the dectin-2 noisiness by comparing two structurally homologous receptors, expressed at the same density, in the same cell line (expanded from a single clone).

To further improve our manuscript and acknowledging that our results are derived in a synthetic model system. We try to make the reader aware of the limited physiological implication of our results. Still, we highlighted, an advantage of using synthetic approaches to dissect precise mechanisms of glycan-mediated signaling as follows:

Line 382: “Finally, it is important to take into consideration that our conclusions came from model cell lines, which were used as a surrogate for cell-type-specific lectin expression patterns of primary immune cells. Human monocytes and dectin-2 positive U937 cells have comparable receptor densities and respond similar to stimulation with zymosan particles (Appendix 1 Figure 6A and B).”

2) One of the most interesting aspects of this work is the fact that the authors consider not just the downstream signaling response, but also directly measure the statistical properties of the first step in the signaling process: namely, receptor binding. I think there is a significant opportunity to understand a critical aspect of cell signaling here in a new and interesting way, and so I have some suggestions for analyses the authors could do that would, I think, really increase the impact of the work.I would be really interested to see what the channel capacity would be for the information flow between ligand concentration and the level of bound receptor. In other words, the authors can calculate the channel capacity between the dose of the ligand that they give the cells, and the distribution of bound receptor that they observe on the cell surface.Looking at the data (e.g. Figure 4A), my intuitive sense is that the channel capacities will be relatively low for that calculation: maybe 2 bits or something like that. Interestingly, that is much, much lower than the theoretically possible value we would expect for receptors expressed at around 1000 copies per cell (which is around 3.2-4 bits, see the Suderman Interface Focus paper mentioned above). This may simply be because the receptors are being expressed exogenously and thus have lower absolute numbers, and broader distributions across the population, than would be observed in a more physiological context. That being said, this is to my knowledge the first actual measurement of the amount of bound receptor on the surface of individual cells. This is thus a great opportunity to calculate the information flow between ligand concentration and bound receptor levels.I should note here that it might be intuitive for the authors to see this as an "upper bound" on the channel capacity for their system; in other words, the channel capacity between the ligand concentration and the bound receptor concentration should set an upper bound for the channel capacity for the entire system. The data processing inequality from Shannon seems, at least on the face of it, to guarantee that. It is important to note, however, that the signaling networks in question do not form a Markov chain; the amount of reporter expression can easily average over the entire signaling history in a particular cell. So, whatever the authors find here, it is not technically an upper bound on the downstream information flow. That being said, it would still be very interesting to know!It should be relatively easy for the authors to just make this calculation based on the data they already have. I do, however, have suggestions for a few experiments that would improve the rigor of this calculation and make the results even more interesting.

We agree on Reviewer #1’s great suggestion on the additional analysis of information transmission in the presence of labelled input. We tried to include our new results regarding this (please see Author response image 1), but we could not fully explain the results due to the following reasons:

The measured labelled inputs signal of the cell at a single time point cannot reflect the total amount of interaction between the ligand and receptor. There are several processes that prevent us from directly connecting detected stimulants and signal initiation from such a snapshot. By the time we record the data, several ligands will be left receptor occupancy, whilst initiating signaling. Additionally, labelled stimulants will be degraded in the lysosome after the endocytosis. Therefore, the channel capacity between the ligand concentrations and bound ligands, in this experiment, cannot be the “upper bound” of the channel capacity between ligand concentrations and GFP expression. Similarly, we see that the channel capacity between bound ligand and GFP expression is lower than the channel capacity between ligand concentrations and GFP expression.

**Author response image 1. sa2fig1:** Using labelled input adds measurable intermediate layer between ligand dose and GFP expression that can be directly related to ligand-receptor bindings. (**A**) Illustration showing response of cell in different doses of labelled inputs. (**B**) Illustration showing information transmittance from labelled input doses to ligand-receptor binding and from ligand-receptor binding to NF-κB translocation (left). And schematics representing measurable information including receptor-bound labelled input of cells (right). (**C**) Response of dectin-1 expressing U937 cells on labelled zymosan in different doses.

We would like to share our experimental results regarding the channel capacity calculation in the presence of labelled input as follows:

Before we introduce our new results regarding this, we want to point out our previous results. As shown in the plot from Appendix 1 Figure 4A, WT cells exhibit non-specific binding to labelled invertase and there is no big difference in channel capacity between WT and dectin-2 calculated from ligand concentration and the level of bound ligands. Hence, it is difficult to delineate the non-specific binding of labeled invertase underlying the binding detected for dectin-2 expressing cells. However, only in presence of dectin-2, we detect downstream signaling and expression of GFP.

Due to the presence of non-specific binding of labeled invertase to dectin-2 positive cells, we had to ensure that the activation of NF-κB via dectin-2 is indeed specific to protein glycosylation (Appendix 1 Figure 1B). In the subsequent experiments, we remove dectin-2 epitopes from the stimulants using α-mannosidase and as a result, could not detect any activation of the GFP expression, although such non-mannosylated protein was still non-specifically recognized by the cells. These results confirm that the overall quantified response is glycan-specific and is not affected by non-specific binding of invertase to U937 cells. The findings are included in Appendix 1 Figure 1B and the main text as follows:

However, for the suggested analysis, we had to circumvent the non-specific binding of invertase to the cells and replaced it with label zymosan as a stimulant. The outcomes of these experiments are the following:

The corresponding channel capacity values in each layer is written in the plots in Author response image 2. The data indicate that the channel capacity between bound ligands to GFP expression is almost always higher than that of between ligand concentration and GFP. Therefore, the downstream pathway connecting bound ligand to GFP expression information has more capability in transmitting information than the receptor itself. In case of dectin-2 stimulation via anti-dectin-2 antibody (Author response image 2), we suspect that lower channel capacity between bound ligand and GFP expression (0.42 bit) than ligand concentration to GFP (0.59 bits) is due to degradation of the APC dye over the time course of 16h stimulation.

**Author response image 2. sa2fig2:** Single cell resolved scattered plot showing distribution of GFP expression and bound-labelled input on WT (A), dectin- (B)1, and dectin-2 (C) cells with various concentrations of labelled-zymosan or dectin-2 antibodies (D).

In the case of channel capacity comparison between ligand concentrations to bound ligand and ligand concentrations to GFP, dectin-1 shows less channel capacity in ligand concentration to bound ligands than the other. Please see more data points shown in Author response image 3.

**Author response image 3. sa2fig3:** Channel capacity values in the presence of labelled input information.

According to Reviewer #1’s comment and general intuition, it is hard to understand that the channel capacity of a full communication channel is higher than a fraction of it. We think there is a dissipation of information in the ligand bound state coming from stimulant digestion or off-rate. Several ligands can bind to dectin-1 and initiate downstream signaling, but some may detach from the receptor prior to internalization, but still initiating signaling.

Note that the channel capacity calculation from bound ligand to GFP expression need extra care on the bias generation since the binning is also applied on the input variable. Therefore, the bias increases more rapidly as increasing the number of binning on both input and output (Author response image 4). On the other hand, insufficient number of binning can underestimate the calculated channel capacity. To address this problem, we employed equal frequency binning. The equal frequency binning partitions the data set such that the individual binning region contains the same number of data points. The equal frequency binning, compared with linear binning, is faster approaches the maximum channel capacity with increasing number of binning (see also Author response image 5).

**Author response image 4. sa2fig4:** Channel capacity calculation including labelled zymosan information. A) Schematic diagram representing both the layers where the corresponding information is measured (rectangular box) and the transmissions of information between layers (colored allows). B-D) Channel capacity values calculated from varied binning number and binning method for dectin-1, dectin-2 and WT U937 cells. The information of individual colors in the plot is given in A). The error bars in the plots represent the 95% confidence interval of the channel capacity value.

**Author response image 5. sa2fig5:** Output binning number and binning method dependence of channel capacity value for experimental dataset. The inset plots show the relative difference of channel capacity value to the maximum channel capacity value in the entire binning range (i.e., from 10 to 1000) of the corresponding binning method.

Taken together, even we could calculate the transmitted information through a receptor bound state, the measured labelled input information does not fully capture the total amount of receptor ligand interactions and therefore we could not quantitatively explain the results. Hence, we provide out experiments and analysis for the Reviewers only. We think implementation of measuring temporal dynamics of labelled ligand binding and unbinding with pH stable fluorophores, can provide actual information of ligand receptor interactions. This is ongoing research in the lab and we again thank the reviewer for his/her input.

Firstly, it is unclear how representative the bound protein numbers are of the actual number of binding events we would have in the cell culture when the cells are exposed to the ligand. This is because the authors have to wash the cells (at least I assume they do that) and then put them through the FACS instrument to measure the fluorescence. It would be really cool of the authors could do a time course to measure how fast the ligand "falls off" of the receptor. As long as the measurements made by the authors occur before significant unbinding has happened, the channel capacity they calculate should be very representative of what actually happened in the culture.

We agree with the reviewer, unbinding of ligand could obscure the data analysis (see also above). In our assay, ligands were always washed before flow experiments. We assume that due to multivalent nature of glycan-lectin interactions the desorption rate for the studied system is very low, which is why the washing step does not affect binding. Our assumption stems from working with these receptors in isolated biophysical assays (e.g. SPR) in previous work (Aretz et al. ACIE 2017). This effect will be even more pronounced if multiple copies of these receptors are present on a cellular surface. Thus, the measured channel capacity is likely to be representative of what is happening in cell culture. We also would like to note that the system presented here is not ideal to investigate dynamics of lectin-glycan interactions. In our follow-up studies, we plan to establish a kinetic microscopy assay to probe such interactions on a single-receptor level. We also plan to use not the end-point reporter, as NF-κB-GFP, but real-time signalling sensors, like ERK-KTR.

The other experiment that would be really interesting would be to repeat the experiment with labeled ligands but using whatever primary cells are known to express dectin-1, dectin-2, etc. In other words, the fluorescence here is coming from the ligand, so there is no need to engineer any reporters into the cell. If the authors see similar distributions of bound receptor on the surface of primary cells, and similar channel capacities between ligand concentration input and the bound receptor as output, that would be very intriguing. In any case, doing this experiment would I think begin to address some of the concerns with physiological relevance raised above. It would also be extremely interesting as a contribution to the field.

According to Reviewer #1’s suggestion we compared labelled zymosan binding and dectin-2 expression level between our transfected U937 cell and primary cultured human monocytes as shown in Appendix 1—figure 6. We found that our model cells express one order of magnitude higher dectin-2 expression relative to the human monocytes (Appendix 1—figure 6B A) and this resulted in the similar labelled zymosan binding trend on both cell types as increasing the zymosan concentration (Appendix 1—figure 6B). We include the following sentence in Appendix 1 Figure 6 and main text, respectively:

Line 382: “Finally, it is important to take into consideration that our conclusions came from model cell lines, which were used as a surrogate for cell-type-specific lectin expression patterns of primary immune cells. Human monocytes and dectin-2 positive U937 cells have comparable receptor densities and respond similar to stimulation with zymosan particles (Appendix 1 Figure 6A and B).”

3) As mentioned in my public comments, another issue in this work is the fact that the authors are relatively reticent about how their calculations are actually performed. While the authors present the relevant equations for calculating the Mutual Information and claim to follow the approach of Cheong et al., the methods are not sufficiently detailed to understand exactly what they have done here. This is particularly important because their approach is clearly not identical to that used by Cheong, since their algorithm for maximizing the MI across input distributions is different. Cheong et al. approach this problem by trying a limited set of possible input distributions and using the one that gives the highest MI, while the authors here use a built-in optimization algorithm in python. It is not clear if this is the only point at which the authors deviate from the approach laid out by Cheong et al., or if they have modified other aspects of the calculation.

According to Reviewer #1’s comment we extended the channel capacity calculation procedure as described in Appendix 2 and 3 for the clear explanation. In the case of modulating input distribution to maximize the channel capacity, we compared the method given in Cheong et al. and our approach as shown in Appendix 2 Section 4.

Mutual information calculation under unimodal and bimodal input distributions. (A) Examples of unimodal input distributions. The parameter s is the standard deviation of the Gaussian function selected from 0.5, 1, 2, 4 and 8. There are 60 cases of input distributions. (B) Examples of bimodal input distribution containing the same s parameters of the unimodal distributions. The number of bimodal combinations of the distribution is 1496. Vertically sorted various unimodal (C) and bimodal (D) input marginal probability distribution by the mutual information yields of the distribution. The probability space for the maximum mutual information given from unimodal (E) and bimodal (F) input distributions.

As shown in Appendix 2—figure 3, we compared between channel capacity values calculated from predefined unimodal/bimodal distribution and that of our optimization approach using built in Python function. We found that our approach gives the same (at least in given significant figures) channel capacity, 1.01 bit, for both cases, if we use bimodal input distribution.

The authors need to explain how they approach the problem of correcting for finite size effects. Do they use the approach of Cheong et al., bootstrapping the data at different levels and then using a linear extrapolation of MI vs. 1/N (where N is the number of data points/cells) to extrapolate to the case of an infinite population size (1/N \to 0).

In the original manuscript we did not consider the bias originated from the finite sample side because the sample number is very high (~150,000). Please see the channel capacities evaluated from experimental dataset (Appendix 2—figure 8).

Channel capacity estimation of experimental data using bootstrapping in various y-binning number. The y-intercept values of the regression line are the estimated channel capacity in the given y-binning number. The number of subsampled data points in each inverse sample side is 30.

The maximum corrected bias using bootstrapping in the entire dataset is less than 0.03 bits. The slopes in the linear regression lines in the plot are inverse proportional to the sample size. Therefore, our relatively bigger sample size does not significantly affect the corrected channel capacity. But we admit that in the case of relatively smaller sample size, using bootstrapping is required and gives more reliable result. Therefore, to retain a generality in channel capacity calculation of this work, we included bootstrapping calculation in all channel capacity estimates. The detailed bootstrapping procedures are shown in Appendix 2 Section 7.

Did they use the same approach to choosing the number of bins to calculate the probability distribution over outputs? Cheong et al. did this by visual inspection, if I recall correctly, looking for a "plateau" in bin numbers where the MI calculated from the data was non-zero but the MI calculated for randomized data was 0. Based on the statements made by the authors, and Figure S6, they seem to have performed both of these steps-otherwise it is completely unclear where the confidence intervals on their MI/channel capacity estimates come from, and how they chose the number of bins to use. Although it is important to note that the randomized controls seem to be missing from Figure S6. Regardless, since the authors have evidently re-implemented the code, they need to explain exactly what they have done.

In the original manuscript, we found appropriate binning number using visual inspection as Cheong et al. did. In the case of bias estimation using randomized dataset, we could not find any noticeable bias generation in the entire binning range from 0 to 1000. But we did not include the data. In the revised manuscript, we employed more rigorous binning method as described below:

Appendix 2—figure 7 shows channel capacities calculated from bootstrapping for various binning and total subsample numbers. In the case of experimental data set (TNF-a stimulation on TNFAR), the total number of samples represent the number of subsampled dataset (without replacement) from the original dataset. The white line in the random dataset represents the 0.01 bits of contour line. This indicates that the bias of channel capacity calculation is not only dependent on the number of output but also the number of measured samples. Note that the smallest number of experimental dataset is around 99,000. Therefore, we can expect that the maximum bias in our channel capacity calculation is less than 0.01 bits in the binning range from 10 to 1000 according to the random dataset. Indeed, as shown in Appendix 2—figure 7D, channel capacity values for subsampled (51151 sample size) experimental and random dataset exhibit stable line in the given binning range.

Accordingly, we decided to choose channel capacity by which the maximum channel capacity value calculated from 10 to 1000 of output binning range. And we included this procedure in Appendix 2—figure 7.

I would suggest that the authors also adopt several improvements to the Cheong et al. approach that have been developed in subsequent works. In particular, Suderman et al. (2017, PNAS 114 (22), 5755-60) implemented a broader range of bootstrap samples across which to do the linear extrapolation to infinite population size, automated the choice of bin sizes, and developed a bootstrap approach to estimating confidence intervals that is much more realistic (and statistically appropriate) than simply using the confidence intervals from the linear extrapolation step. The authors can refer to the Suderman paper on how this was done; it should be relatively easy to implement these improvements, if they have not been done already.

According to Reviewer # 1’s suggestions, we have implemented additional statistical methods to improve the channel capacity estimates of the work as follows: We implemented bootstrapping method, and by using linear regression we extrapolated channel capacity value at infinite sample size (Appendix 2 Figure 6). On the other hand, the confidence intervals, having less than the minimum value of significant figure, are not present in this work. Instead, we compared different outcomes using non-parametric statistical test as Reviewer #1’s suggestion from comment #2.

In the case of binning size selection, we decided to choose channel capacity by which the maximum channel capacity value calculated from 10 to 1000 of the output binning range as described in Appendix 2—figure 7.

I appreciate the author's innovation of using an optimizer in python to maximize the MI. I would expect this actually produces higher "channel capacities" than the previous approaches mentioned above. That being said, it is unclear how this optimization algorithm works. The authors may expect readers to go to the python documentation, but there are problems there. For one, the algorithm implanted in python might change, so the documentation may describe an approach that is different from the one the authors actually used to perform their calculations. How various packages are implemented in python can change from time to time, and something so generic as "optimize" may be changed as better optimization algorithms become available in the future. So the authors should describe exactly what they did. Also, readers are, in general, not going to go to python documentation to understand the methods of a paper like this, nor should they have to. The authors should describe what the algorithm is doing as currently implemented, in terms that readers can readily understand.

We agree to Reviewer #1’s comment regarding the lacking explanation of mutual information maximization procedures. The explanation should be more general and independent. Therefore, we have corrected Appendix 2 Section 3. describing mutual information procedures as follows:

“Finding input weighting values, w(i), that maximize the mutual information subject to ∑iinputw(i)Px(i)=1 and 0≤w(i)Px(i) is a non-linear optimization problem. Since, the direction of the gradient of mutual information is the same as that of those two constraints at the minimum, using Lagrange multiplier method, one can restated the functions as Lagrangian L(wi,λ,σ)=MI(input;output|w(i))−λ[∑iinputw(i)Px(i)−1]−σ[w(i)Px(i)], and find out weightings using numeric approach (Kraft D. 1988. A Software Package for Sequential Quadratic Programming. Wiss. Berichtswesen d. DFVLR). We used Sequential Least Squares Programming (SLSQP) provided by SciPy Python library (scipy.optimize.minimize, SciPy 1.7.3) to find out the optimizing input weighting values.”

Finally, the authors should also perform the calculation using the exact same distributions used by Cheong et al. and Suderman et al. This will allow us to compare their results more directly to the results from those previous authors, as well as allow us to determine if the approach that they used results in similar channel capacities to the optimization algorithm used here. While I expect the optimization algorithm employed by the authors will yield larger numbers for the channel capacity (and thus better estimates, since the channel capacity is formally a supremum), it would be extremely useful if the authors checked this.

According to Reviewer #1’s suggestion, we compared the channel capacities calculated from predefined bimodal and unimodal input distribution with our systematic approach as described in Appendix 2—figure 3 and Appendix 2 Section 4. We found that there is no noticeable channel capacity difference between those two methods in our significant figure range.

4) Another issue related to the calculation of the MI itself is the fact that the authors use logarithmically spaced bins, rather than linear bins, to perform the calculation. The authors describe this approach as if the matter of how to construct the bins is simply a matter of personal preference or convenience for the calculation, and choose logarithmic bins because they give somewhat larger values for the channel capacity. For one, this is not surprising; as is often the case, the FACS data obtained by the authors has a log-normal-like character; in other words, the authors see normal-ish distributions on a log scale (e.g. Figure 2A). As such, it is natural that using logarithmic bins will result in higher channel capacities, because it will tend to equalize (as much as possible) the effective number of observations within the bins.While this may seem reasonable at first, it is actually, in my view, hard to justify. The reason is that the choice of how to generate the bins is not arbitrary, but actually corresponds to a very strong statement regarding what the authors actually consider the relevant "output" of the system to be. Using logarithmic bins is obviously equivalent to first taking the log of the data, and then using linear bins on that log-transformed data (Figure S2A should make this fact abundantly clear). The authors are thus envisioning that the channel is not a channel whose input is ligand concentration and output is GFP level, but rather a channel where the input is the ligand concentration and the output is the log of the GFP level. This is not an arbitrary decision, but rather one that has real consequences for how we construe the calculation in a physiological context. If the output is the log of the GFP level, that means that the cell is sensitive not to linear changes in protein concentration, but rather fold-changes.For certain transcription factors, authors have argued that it is actually this kind of relative, fold change that matters (see Lee et al. (2014) Mol Cell 53 (6) 867-79). I would argue, however, that the conclusions of that work are hardly so robust as to suggest that, for every input-output transcriptional regulatory system within cells, it is the fold change of the "output" protein, rather than its linear change, that matters to the cell. Certainly, if the output is an enzyme, a transporter, or a protein that performs a whole host of other functions, then it is natural to assume that the appropriate output is the protein concentration itself, not the fold change in protein concentration.If the authors can argue that every single gene whose transcription is controlled by NFκB has a fold-change impact on its downstream function, then I could see a strong argument being made for logarithmic bins. As it stands, however, I think the most natural way to interpret the output of the channel is on a linear scale, and I would say the authors should focus on that, rather than a logarithmic scale. If the authors want to include their logarithmic calculations in this work, I would suggest moving them to the supplement, and making clear the fact that such a calculation construes the output as having a fold-change impact on whatever is downstream of the protein level measured by the authors.

We understand Reviewer #1’s comment. Please see the public comment #3 and our answer including Author response image 5. We found whether we choose logarithmic, linear or equal frequency binning, there is no significant difference in the calculated channel capacity values if the binning number is more than 200. The binning method, however, should not be arbitrary but has to have strong relationship with the context. Therefore, in the revised manuscript, we used linear binning and corrected all the channel capacity values accordingly. Please note that the new calculation does not affect our conclusion.

5) In this work, the authors focus on the measurement of GFP levels at a certain time. As written, it is not clear if the authors are considering the steady-state response of the cells that they are treating, a peak response, or something else. As far as I can tell, the authors stimulate the cells for 16 or 13 hours, depending on the ligand in question. I am not sure how these numbers were chosen-did the authors look at the GFP expression dynamics and choose this number based on those measurements? From an information-theoretic standpoint, it is best if the results are based on steady-state protein levels, but the author's don't seem to explain their rationale for choosing the time points that they choose. If the GFP expression levels do not reach a steady-state, it would be great if the authors had a particular reason for choosing the times they choose. I would also suggest that the authors consider looking to see if their channel capacity estimates are robust to the time they choose by looking at other time points and calculating the channel capacities to see if they get similar results. I am not sure what times to choose, but perhaps 24 hours or 36? It is hard to say without looking at the average dynamics over time, so providing that kind of data would be extremely helpful.

Thank you for bringing this to our attention. We noticed that the stimulation time written in Materials and methods part is not correct. We stimulated 16h in every experiment except TNF-a stimulation as described in the original main text. The time at which the mean value of the GFP expression level reaches its maximum was determined as the stimulation time. As shown in (Appendix 1—figure 2), U937 cells exhibit maximum mean GFP response at around 12 hours of TNF-a stimulation (50 ng/mL) while the expression level stay in steady state. And at around 20 hours later, the GFP expression level decreases to the basal level. Therefore, we selected 13 hours as the stimulation time if the TNF-a is the stimulant. For the other stimulants, we used 16 hours as the stimulation time according to the maximum GFP response. Author response image 6 shows the channel capacity estimate of dectin-2 expressing U937 cell using mannan as the stimulant at different time points. We include these data into our manuscript as follows:

Line 168: “Note that we choose the stimulation time, the period of incubation time of the cell with the input ligands, as the time point when GFP response and channel capacity reaches the maximum and steady state value (Appendix 1 Figure 2A and B).”

**Author response image 6. sa2fig6:** Channel capacity of TNFαR channel to TNF-α stimulant for various stimulation time. Error bars indicate 95 % confidence interval.

Also, the authors should acknowledge that a large body of literature has emerged that makes the claim that it is not the individual time points that matter, but rather the entire time series of the response that should be construed as the output. The authors can refer to the Zhang et al. paper in Cell Systems, the paper by Selimkhanov et al. (Science (2014) 346 (6215) 1370-3), papers by the Hoffman group (notably Tang et al. (2020) Nat Commun 12 (1) 1272), and a host of others. I am not suggesting that the authors adopt this worldview; there are actually serious mathematical and conceptual issues with construing the output of a communication channel to be a function like a time series. But, this idea is out there in the literature, and the authors should address this point and discuss how looking at GFP time series, rather than individual time point measurements, might yield different (and undoubtedly higher) channel capacity estimates.

We appreciate Reviewer #1’s suggestion for discussing channel capacity calculation from dynamics datasets. We understand there are lots of variability of selecting input and output structure. For example, one might use oscillating ligand concentration as an input and collect temporal NF-κB translocation into nucleus as an output (Kellogg, R. A. & Tay, S. Cell 160, 381–392 (2015)), which seemingly, even the authors did not consider information theory, yields more higher channel capacity than static input. Indeed, there are a lot of ways of selecting input and output for channel capacity calculation in the same signaling pathway. In this work, we were concerned with the information transmission via carbohydrates via lectin signaling pathways in a cell population level at a fixed time point. And we agree that describing channel capacity measurement using single cell and time resolved cellular output brings more broader insight on this work. This is ongoing research in the lab. We added following sentence in the Discussions and Conclusions part as follows:

Line 174: “In addition, this channel capacity can be further increased if one can measure the temporal evolution of output dynamics instead of static output dataset (Selimkhanov et al., 2014).”

6) As a final, rather important but also rather technical point, the authors in Figure 3C do statistical tests on their channel capacity estimates, comparing dectin-1 and dectin-2 cells to those that express dectin-1 and dectin-2 together. The issue here is that they use a t-test to estimate statistical significance, but it is really unclear where the dispersion (e.g. error bars or standard deviations) in the channel capacity estimates come from. I expect, since the authors day "n = 9" in the legend, that they split their data into 9 groups (maybe 9 sets of cells whose results were collected in different FACS runs or something?), performed the channel capacity estimate on each independently, then used the means and standard deviations in those estimates to do the t-test. That is completely made up by me, however; the authors really need to explain where these error estimates (and their +/- X values for channel capacity estimates in the various tables) actually come from.In any case, the t-test of course only works if the data in question is drawn from actual Gaussian distributions. This does not mean that the distributions "look Gaussian" or "seem okay;" in the application of a t-test, you make extremely strong assumptions that only real honest-to-goodness Gaussian distributions actually satisfy. So the data really needs to actually be drawn from a Gaussian distribution for the results of the test to be interpreted appropriately.My first suggestion here is that the authors need to use a non-parametric statistical test. That is, unless there is a true, theoretical reason to expect that the data will be drawn from a Gaussian (I know of no such reason, but perhaps the authors have one). The Wilcoxon rank-sum test was made for problems like this one, so I would suggest the authors use that.A larger problem here, however, is that taking n = 9 (however it was done in the end) is not going to give a great estimate of the uncertainty in the channel capacity estimate. I would instead suggest that the authors merge their data into one data set (unless there are some kind of batch effects that prevent them from doing that) and then using a bootstrap/resampling approach, as in the Suderman et al. PNAS paper, to estimate confidence intervals and generate distributions for statistical tests. These bootstraps can be done hundreds or thousands of times, allowing for even more non-parametric tests (like permutation tests) to be used to estimate statistical significance.This is honestly a fairly minor point, because nothing terribly important in the conclusions of the paper rest on the statistical tests here. But, if the authors want to argue that there are real differences between dectin-1, dectin-2 and dectin-1+dectin-2 cells, they should do these statistics in a more non-parametric way.

Since it is unclear whether the channel capacities are distributed Gaussian distribution across different batches of experiment, we have changed statistical test from t-test to Wilcoxon rank-sum test, a non-parametric statistical test, if we compare channel capacities in the manuscript.

Figure 2B and D include the significances from Wilcoxon rank-sum test, indicating the preserved statistical significance (*p>0.05, **p>0.01) that we had in the original manuscript.

In the revised manuscript, we do not compare the other pairs of receptors due to the possible difference in downstream molecule number of U937. And also please note that the pooling the dataset that measured in different dates usually generate high bias in the channel capacity since the baseline of the GFP signal is significantly different due to the variation of laser condition of flow cytometer day-to-day.

Reviewer #2 (Recommendations for the authors):I think this paper would be greatly improved if the authors were more precise and detailed in the description of their capacity calculation. To address this, it's likely the authors were following the method from Cheong et al., but some of those details should be described here.

We thank Reviewer #2’s suggestion of improving the description of channel capacity calculation. Therefore, in the revised manuscript we added the following contents in Appendix 2.